# Stability evaluation and potential failure process of rock slopes characterized by non-persistent fractures

Wen Zhang[1], Jia Wang[1], Peihua Xu[1], Junqing Lou[2], Bo Shan[2], Fengyan Wang[3], Chen Cao[1], Xiaoxue Chen[1], Jinsheng Que[4]

[1]College of Construction Engineering, Jilin University, Changchun, 130026, China
[2]Northeast Electric Power Design Institute CO., LTD. of China Power Engineering Consulting Group, Changchun, 130026, China
[3]College of Geo-Exploration Science and Technology, Jilin University, Changchun, 130026, China
[4]CCB Cost Engineering & Consulting Co., Ltd., Beijing, 100000, China

*Correspondence to*: Peihua Xu (peihxu@126.com), Chen Cao (ccao@jlu.edu.cn)

**Abstract.** Slope failure, which causes destructive damage and fatalities, is extremely common in mountainous areas. Therefore, the stability and potential failure of slopes must be analyzed accurately. For most fractured rock slopes, the complexity and random distribution of structural fractures make the aforementioned analyses considerably challenging for engineers and geologists worldwide. This study aims to solve this problem by proposing a comprehensive approach that combines the discrete fracture network (DFN) modeling technique, synthetic rock mass (SRM) approach, and statistical analysis. Specifically, a real fractured rock slope in Laohuding Quarry in Jixian County is studied to show this comprehensive approach. DFN simulation is performed to generate non-persistent fractures in the cross section of the slope. Subsequently, SRM approach is applied to simulate the slope model using 2D particle flow code software (PFC2D). A stability analysis is carried out based on the improved gravity increase method, emphasizing the effect of stress concentration throughout the formation of the critical slip surface. The collapse, rotation, and fragmentation of blocks and the accumulation distances are evaluated in the potential failure process of the rock slope. 100 slope models generated with different DFN models are used to repeat the aforementioned analyses as the result of a high degree of variability in DFN simulation. The critical slip surface, factor of safety, and accumulation distance are selected by statistical analysis for safety assurance in slope analysis and support.

**Keywords.** Fractured rock slope, Synthetic rock mass, Statistical analysis, Stability evaluation, Potential failure

## 1 Introduction

Rockslides are common geological hazards in mountainous areas. This phenomenon seriously threatens human lives and properties worldwide every year. Therefore, the analyses of rock slopes, especially their stability and failure process, are necessary for civil and mining engineering. Generally, rockslides are controlled by discontinuities (such as beddings, faults,

and structural fractures). Discontinuities in rock slopes were formerly assumed in slope analysis as through-going and fully persistent planes. However, many researchers have suggested that discontinuities are generally not fully persistent (Eberhardt et al., 2004; Scavia, 1995; Terzaghi, 1962) and that failure surface is a combination of preexisting non-persistent discontinuities and newly propagated cracks (Brideau et al., 2009; Einstein et al., 1983; Frayssines and Hantz, 2006; Lajtai, 1969a; Zhang et al., 2017). Non-persistent fractures are highly complex and come in various sizes and properties, and their locations and characteristics are difficult to determine. Thus, the influence of non-persistent fractures on slope analysis poses a great challenge (Fan et al., 2015; Wasantha et al., 2014).

Several researchers showed that non-persistent fractures play vital roles in the stability and failure process of fractured rock slopes (Gao et al., 2017; Huang et al., 2015; Jiang et al., 2015; Li et al., 2009; Zhang et al., 2015; Zhou et al., 2017). However, the non-persistent fractures in these works were artificially specified by researchers as having the same sizes and orderly arranged locations. The complexity and random distribution of non-persistent fractures in rock slopes should be considered in reflecting and modeling the stability and failure process of real rock slopes in nature.

According to a major research work on fractured rock masses, the discrete fracture network (DFN) modeling technique maximizes the use of discontinuity data from exposed surfaces and can become the best option for simulating realistic fractured rock masses (Bonilla-Sierra et al., 2015; Chen, 2001; Elmo et al., 2013; Pine et al., 2006). By coupling the DFN technique with continuum, discontinuum, and hybrid modeling approaches, synthetic rock mass (SRM) models can be set up to investigate the mechanical properties of fractured rock masses (Elmo and Stead, 2009; Mas Ivars, 2010; Pierce et al., 2007). SRM approach has been widely used to determine the representative elementary volume size of a fractured rock mass (Esmaieli et al., 2010), reproduce rock mass properties and behaviours (Mas Ivars et al., 2011), and simulate fracture propagation in a fractured rock mass (Zhang and Stead, 2014). SRM models have been primarily used to simulate failure and deformation of fractured rock slopes (Bonilla-Sierra et al., 2015; Elmo et al., 2013), simulate hydraulic fracturing in naturally fractured reservoirs (Damjanac and Cundall, 2016), and estimate rock mass strength, fragmentation and micro seismicity in caving mines (Lorig et al., 2017). DFN simulation included in SRM modeling program presents a significant variability, which means numerously possible realizations of 2D fracture systems exist given specified input parameters (Pine et al., 2006; Zhang et al., 2020a). The results of SRM model analyses that incorporate different DFN models may vary significantly (Elmo et al., 2018; Zhang et al., 2012). Therefore, rock slope analyses based on only one DFN model can lead to erroneous results (Elmouttie and Poropat, 2014; Mas Ivars, 2006). A statistical analysis based on a large number of DFN models may reduce the aforementioned errors and provide reasonable results for rock slope analysis and support (Ferrero et al., 2016; Zhang et al., 2013; Zhang et al., 2017).

The current study proposes a comprehensive approach that combines several well-established methods to conduct a stability evaluation and failure process analysis of a fractured rock slope in Tianjin City, China. First, 100 DFN models are generated on the basis of the fractures collected in the field. Second, slope models are constructed using SRM approach. Third, the improved gravity increase method is employed to determine the stability of all slope models generated with 100 DFNs. Fourth, the potential failure processes of the fractured rock slope models are simulated. Fifth, the final critical slip

surface, factor of safety, and accumulation distance are determined on the basis of the statistical analysis of 100 slope models generated with different DFN models.

## 2 Study area and data acquisition

### 2.1 Study area

A rock slope in Laohuding Quarry, which is located north of Jixian County, Tianjin, is analyzed in this study (Fig. 1). The

quarry was originally used for mining stromatolites, and this activity seriously damaged vegetation. After the mining area was closed, many steep slopes with internally developed discontinuities were formed. These slopes may develop rock falls and pose potential threats to people and nearby equipment. Understanding whether rock slides will happen requires careful geological investigations.

The Laohuding Quarry area is in a low-mountain terrain, characterized by higher elevation to the north than to the south.

The highest and lowest altitudes of the quarry area are 160 m and 60 m, with a relative elevation of 100 m. A majority of monoclinal mountains striking south–north exist in this area. The average slopes of the mountains in the east and west of the quarry area are 25 ° and 30 °, respectively (Fig. 1b).

This region has a continental climate with an average annual precipitation of 770.20 mm. The flood season is from June to August, and it accounts for 77.3 % of the annual precipitation. During this period, slope failures are frequently triggered.

The average annual evaporation is 1867.30 mm, which is 2.42 times the amount of precipitation. These conditions indicate that the region is arid.

The lithology in this area is characterized by limestone of the middle-upper Proterozoic erathem, which exhibits a powder crystal–mud crystal structure. The limestone is moderately weathered, and the karst phenomena are scarcely visible due to low precipitation and groundwater shortage. Bedding planes, weak interlayers, faults, folds and shear zones are not

observed. This area is tectonically stable according to historic seismicity data. Non-persistent discontinuities are randomly and widely developed in outcrops (Fig. 1c). Therefore, it is the non-persistent discontinuities (fractures) that control the slope stability and potential failure process.

### 2.2 Data acquisition and DFN generation

The investigated rock slope is 20 m high and is oriented at a trend of 200 ° with an approximately vertical angle (Fig. 1c).

Fractures in the exposed surface with trace lengths smaller than 1.5 m are widely distributed and difficult to record. Therefore, these fracture are not considered when performing fracture data collection for 2D DFN simulation. Nevertheless, the effect of these fractures on the rock mass strength is considered, which is explained in Sect. 3.1. The characteristics (such as orientation, trace length, spacing, roughness, aperture, filling, and termination) of fractures with trace lengths larger than 1.5 m were systematically surveyed using the sampling window method (Kulatilake and Wu, 1984). The rectangular

sampling window was measured 62 m long and 6m high. Exactly 169 fractures were collected, and the 2D trace map is

shown in Fig. 2. The mechanical properties of the different rock mass structures vary, and thus, an evaluation of the homogeneity of the fractured rock mass was conducted (Zhang et al., 2011). The result shows that this region can be considered as a statistically homogeneous area. The fractures can be divided into three sets using the method proposed by Chen et al. (2005), as shown in Figure 3. Fractures in set 1 with an average orientation of 39.5 °/87.3 ° (dip direction/dip angle) are rare. Fractures in sets 2 and 3 with an average orientation of 307.4 °/44.7 ° and 110.2 °/31.7 °, respectively, host the most fractures and constitute the dominant sets (Table 1). The computed Fisher constants ($K$) for three fracture sets are 17.1, 10.3, and 9.1, which imply the high dispersion of fracture orientations (Fisher, 1953; Priest, 1993).

The sampling window method features two main limits of orientation bias and trace length bias. The measured trace lengths bias occurs due to the following: the size of the sampling window is usually smaller than that of the outcrop. When the trace length is larger than the sampling window, the actual trace length cannot be totally measured, i.e., only the part of the trace length inside the sampling window can be measured. Specifically, the measured results can be divided into three types: a) only one end of a fracture trace is measured, (b) both ends of a fracture trace are measured, and (c) no end of a fracture is measured. The mean value and probability density function (PDF) of the corrected trace lengths for each fracture set can be determined according to the methods proposed by Kulatilake and Wu (1984) (Table 1). Orientation bias occurs because the fractures with small intersection angles between the fractures and exposed rock surface are more possibly collected in the field than those fractures with large angles (Nie et al., 2020). In the present study, a 2D analysis was performed. The cross section normal to the exposed surface was used to perform the 2D stability analysis and investigate the potential failure mechanism of the rock slope. The dip directions of fractures in set 1 are similar to that of the cross section, implying fractures acted as the surface of separation, which will not influence the results of stability analysis. Therefore, a substantial portion of the fractures of set 1 (fractures intersected by the cross section with an angle smaller than 20 °) were artificially deleted prior to the stability analysis.

We derived the characteristics of the fractures (location, orientation, trace length, and density) in the cross section from those features measured across the exposed surface. Subsequently, the 2D fracture traces were generated using Monte Carlo simulation by synthesizing the aforementioned characteristics (Li et al., 2020). The specific processes are clearly described in the research of Zhang et al. (2017) and are briefly introduced as follows. The locations of the fractures were assumed to follow Poisson's distribution. We rotate the slope 20 ° so that the slope strikes in the NS direction and assume the fracture frequency measured along the mean normal vector direction of fracture set $i$ is $\lambda_i$, and the acute angle between this direction and NS direction is $\eta_i$. The fracture frequency along the line parallel to the strike of the outcrop plane is $\lambda_i\cos\eta_i$ (Priest 1993), and the cross section plane is $\lambda_i\sin\eta_i$. The fracture frequency of the latter is $\tan\eta_i$ times that of the former, and $P_{21}$ (2D fracture intensity) follows this result according to the concept of the integral. The 2D orientation of a fracture is reflected by its trace gradient $k$, which can be expressed as $\sin\alpha/\cot\beta$ (where $\alpha$ and $\beta$ are the dip direction and dip angle of a 3D fracture, respectively). Empirical distribution was followed to generate the 3D fracture orientations based on the fracture orientation frequency, and then $k$ in the cross section can be determined. The results of Zhang et al. (2017) revealed that the mean [$E(ch^2)$] and variance [$V(ch^2)$] values of the trace length square are constant. $E(ch^2)$ and $V(ch^2)$ of the collected fracture traces

in the cross section can be determined according to those features on the exposed surface. The rooted numbers result in the trace lengths in the cross section. The cross section of the investigated rock slope was 20 m high, and the smallest mean dip angle of the three fracture sets was 31.7 °. Thus, the length of the cross section was 30 m [20 × cot (31.7 °)]. Finally, we used Monte Carlo simulation to merge the aforementioned parameters to generate the 2D DFN model in a 30 m × 20 m cross section.

More than one DFN can be generated on the basis of the aforementioned statistical fracture data, including the distribution type of fracture locations (Possion's distribution), 2D fracture intensity ($P_{21}$), the distribution type of fracture orientation (empirical distribution), and the mean [$E(ch^2)$] and variance [$V(ch^2)$] values of the trace length square. For example, Fig. 4 exhibits four DFNs with the same statistical fracture data, but fracture characteristics, such as locations, dip angles, trace lengths, are different from one another. Therefore, the rock slope analyses based on only one DFN model can

lead to wrong results (Mas Ivars, 2006; Xu et al., 2014). To solve this problem, we generated and verified the validity of numerous DFN models by applying the procedure in Fig. 5. Initially, we set lines along the strike of the exposed surface and fractures in the exposed surface can be intersected with these lines. A series of intersection points can be obtained for an individual line. Subsequently, the same procedure was performed to the generated DFNs, i.e., we intersected the DFNs using lines along the strike of the cross section and consequently generated a series of intersection points for an individual network.

We compared these two sets of intersection points by using their probability density curves. When the results were identical to one another, the DFNs generated in the cross section were proved reasonable and can be selected. Finally, we selected 100 reasonable DFN models to construct different slope models and conduct stability analysis and potential failure process simulation.

## 3 SRM model for rock slope analysis

SRM approach is used to construct fractured rock slope models. In the present study, the SRM model used was a combination of two well-established models, namely, the bonded-particle model (BPM) in PFC2D and the DFN model (details are available in the research of Pierce et al. (2007)). SRM approach is widely used to reproduce the mechanical properties and behaviours of fractured rock masses, simulate the fracture propagation and brittle failure of fractured rock masses, and perform stability analysis of fractured rock slopes.

### 3.1 Parameter determination for SRM model

The SRM model in PFC2D is defined by many parameters, such as particle contact modulus, particle normal/shear stiffness ratio, and parallel bond modulus. These parameters cannot be directly identified via laboratory and field experiments. Therefore, the parameters of the model should be predetermined according to the macroscopic characteristics of rock slopes. In particular, the input parameters of the BPM of the rock mass between the large fractures and DFN model for fractures

should be ascertained and quantified separately.

Field investigation only collected the large fractures (fractures with trace length larger than 1.5 m) which have been considered in DFN model; thus, the effect of small fractures should be included in the BPM of the rock mass between the large fractures. The BPM parameters are calibrated against the macro-properties of the rock mass, such as uniaxial compressive strength ($\sigma_c$), elastic modulus ($E$), Poisson's ratio, friction angle, and cohesion. These macro-properties of the rock mass are the results of comprehensive action of small fractures and intact rock material. Rock mass failure is generally shear failure and thus the shear strength of the rock mass needs to be focused on. $\sigma_c$, $E$ and Poisson's ratio have little effect on the shear strength of the rock mass. Therefore, $\sigma_c$, $E$ and Poisson's ratio of the rock mass are assumed the same as those of intact rock material, which are obtained by laboratory uniaxial tests (Table 3). Friction angle and cohesion directly control the shear strength of the rock mass; thus, friction angle and cohesion of the rock mass should be calculated by considering the effect of small fractures and intact rock material. Eq. (1) is originally used to calculate the shear strength of non-persistent fractures which is considered as the linear combination of rock bridges and persistent fractures (Lajtai, 1969b; Shang et al., 2018). The shear strength of non-persistent fractures in Eq. (1) refer to that of the shear failure surface of the rock mass along the shearing direction. Therefore, Eq. (1) can be used to approximately estimate the shear strength of the rock mass. Then, the cohesion and the tangent of friction angle of the rock mass can be calculated by Eqs. (2) and (3).

$$\tau = c_e + \sigma \tan \varphi_e = \left[K_L \cdot c_f + (1 - K_L) \cdot c_R\right] + \sigma\left[K_L \cdot \tan \varphi_f + (1 - K_L) \cdot \tan \varphi_R\right] \tag{1}$$

$$c_e = K_L \cdot c_f + (1 - K_L) \cdot c_R \tag{2}$$

$$\tan \varphi_e = K_L \cdot \tan \varphi_f + (1 - K_L) \cdot \tan \varphi_R \tag{3}$$

where $\tau$ and $\sigma$ represent the shear strength and normal stress of the rock mass between the large fractures; $c_e$ and $\varphi_e$ are the cohesion and friction angle of the rock mass between the large fractures; $c_f$ and $\varphi_f$ are the cohesion and friction angle of small fractures; $c_R$ and $\varphi_R$ are the cohesion and friction angle of intact rock material; $K_L$ is the linear persistence in the shearing direction of the rock mass between the large fractures.

Laboratory uniaxial and biaxial compression tests of intact rock material obtained the values of $c_R$ and $\varphi_R$, which are 12.25 MPa and 25 °, respectively. Direct shear tests of fractures indicated the value of $\varphi_f$ is equal to 18 °. Field investigation demonstrated that no fillings existed in fractures, implying that the cohesion of small fractures is equal to zero (i.e., $c_f = 0$). The linear persistence is defined as the ratio of fracture trace lengths (only the trace lengths of small fractures in the rock mass between the large fractures are considered in the present study) and the total length of coplanar given line (Shang et al., 2018; Zhang et al., 2020b). In the present study, several lines along different directions are set in the rock mass between the large fractures and then the linear persistence is measured. The value of the largest linear persistence measured is around 50%. The direction of the largest linear persistence is considered as the shearing direction of the rock mass between the large fractures due to the lowest shear strength in this direction. Therefore, $K_L$ is equal to 0.5. Substituting the aforementioned parameter values (including $c_R$, $\varphi_R$, $c_f$, $\varphi_f$ and $K_L$) into Eqs. (2) and (3), we can obtain the cohesion and friction angle of the rock mass between the large fractures are equal to 6.125 MPa and 21.58 °, respectively (Table 3).

The specimens (the height-to-diameter ratio is 2:1) for the numerical uniaxial and biaxial compression tests are set up to reproduce the macro-properties of the rock mass between the large fractures in Table 3. In PFC2D, a rock material is considered an assemblage of bonded rigid circular particles, and particle size distribution in rock models dramatically influences modeling behaviour (Mas Ivars et al., 2011). Therefore, determining the particle size of a numerical specimen remains a challenge. Generally, the ratio between the maximum and minimum radii of particles is 1.66 (Potyondy and Cundall, 2004). A small particle size is indicative of accurate simulation results. However, the number of particles of rock models increases with a decrease in particle size. A large amount of particles will result in a long calculation time and low computational efficiency. In the present study, the trial-and-error method was used to find a best balance of reasonable results and high computational efficiency. Specifically, we repeatedly changed the particle sizes and checked the simulation result and the time took. Particles with radii between 0.05 and 0.083 m were finally selected to fill rock specimens.

Different macro-properties are influenced by different parameters in PFC2D. Specifically, $E$ is mainly controlled by several parameters, including particle contact modulus ($E_c$), particle normal/shear stiffness ratio ($k_n/k_s$), parallel bond modulus ($E_b$), and parallel bond normal/shear stiffness ratio ($k_{nb}/k_{sb}$); Poisson's ratio is influenced by $k_n/k_s$ and $k_{nb}/k_{sb}$; $\sigma_c$ is influenced by tensile strength and cohesion of parallel bonds. Finally, the coefficient of friction is calibrated using the results of biaxial tests (Bahaaddini et al., 2013). The values of the aforementioned parameters are empirically assigned in advance, and then the numerical uniaxial and biaxial compression tests are carried out. When the macro-properties of the numerical tests correspond to the results from the laboratory tests, the parameters are considered reasonable. Otherwise, the parameters are adjusted until the rock specimens have the same macro-properties as the rock mass (Park and Song, 2009; Yang et al., 2006). The calibrated parameters of particles and bonds are listed in Table 2. The macro-properties of the rock specimens for the numerical tests and those of the rock mass between the large fractures are listed in Table 3.

In PFC2D, a smooth joint (SJ) model is commonly adopted to simulate the large fractures in DFN. In this model, two particles that lie on the opposite sides of the intended fracture plane can overlap and interpenetrate each other instead of moving along their perimeters, thereby reproducing the real physical and mechanical properties of fractures (Bahaaddini et al., 2013). The SJ parameters are normal stiffness ($\bar{k}_{nj}$), shear stiffness ($\bar{k}_{sj}$), and coefficient of friction ($\mu_j$). To determine these parameters, we built a numerical specimen (with a width-to-height ratio of 1:1) for the direct shear tests and normal deformability tests to simulate macro-properties, including shear stiffness, normal stiffness, and friction angle. The SJ parameters can be obtained as long as the test results approximate those of the laboratory tests. Specifically, $\bar{k}_{nj}$ was obtained by the numerical normal deformability tests; $\bar{k}_{sj}$ and $\mu_j$ were determined by the numerical direct shear test. Table 4 lists the values of the SJ parameters and the results of the normal deformability and direct shear tests of the numerical and laboratory tests. The specific calibration procedures are complex and are thus not introduced in this paper in detail. Additional details about the calibration can be found in the works of Bahaaddini et al. (2013), Cheung et al. (2013), and Duan et al. (2016).

## 3.2 Model generation

The investigated rock slope with a relatively low height is located on a ground surface with a relatively flat terrain. The study area is dry, crustal movement is not obvious, and no active fault exists nearby (Sect. 2.1). Thus, the effects of ground stress, water, and earthquake on the slope analysis were not considered in the current work.

The size of the slope section is 20 m (height) $\times$ 30 m (length). The bottom and right sides of the slope section were expanded by 10 m as the boundary section, which aims to avoid boundary effect and does not affect the slope stability (Fan
et al., 2004). Ultimately, the size of the SRM model was determined to be 30 m (height) $\times$ 40 m (length) (Fig. 6). The upper boundary of the model was free. Moreover, the left, right, and bottom boundaries were assumed to be smooth rigid walls.

The particles with the same radii (0.05 to 0.083 m) as those reported in Sect. 3.1 were applied to fill the 20 m $\times$ 30 m slope section. Considering the small effect of boundary section on the slope analysis, we filled the boundary section with particles with larger radii (0.1 to 0.15 m) to improve computational efficiency (Fig. 6). A total of 48,947 particles were
generated, and the parameters presented in Table 2 were adopted for the bonds and rock particles. Notably, the values of three shear strength parameters (including tensile strength and cohesion of parallel bond and friction coefficient of particles) has been reduced by half according to Sect. 3.1.

Gravity was applied to the model, and the model was calculated (cycled) until the particle assemblage reached an equilibrium state (i.e., the unbalanced forces reached the required standard of $10^{-6}$). Then, an embedded scripting language in
PFC, i.e., FISH, is used to write user-defined functions for extending the functionality or adding user-defined features in PFC. In the present study, we used the FISH functions to add the DFN into the model of the slope section by reading the location data of fractures. Subsequently, the SRM model composed of the BPM and DFN is established (Fig. 6, with the DFN in Fig. 4a as an example). The SJ parameters presented in Table 4 were adopted for the fractures. This process ignored the stress concentration at the tips of the structural fractures generated by tectonic stress, which was considered reasonable in
this study since the investigated slope is characterized by the low stress conditions and the stress concentration was intensely reduced after the long-term stability of the rock slope.

The slope was formed within one operation. Therefore, the left boundary (smooth rigid wall) was removed to simulate a one-time excavation of the slope.

## 4 Stability analysis

### 4.1 Determining the factor of safety

To trigger instability, an improved gravity increase method is used in this study. The improved gravity increase method was proposed by Meng et al. (2015). This method leads to the failure of a slope in PFC2D by slowly increasing gravity acceleration and reducing the friction coefficient of particles while keeping other parameters constant. Notably, the amplitude of reduction of the friction coefficient is the same as that of the increase in gravity acceleration. In this way,

the resisting force can be fixed and therefore the factor of safety is directly reflected by the driving force. When the slope model is in a limit equilibrium state, i.e., fractures in the slope are propagated and coalesced until a through-going slip surface (i.e., the critical slip surface) is initially formed, the factor of safety $F$ can be defined as the ratio of the gravity acceleration in the limit equilibrium state ($g'$) to that in the initial state ($g$), i.e., $F= g' / g$. Taking the model in Fig. 6 as an example, we calculate the factor of safety by using the calculation procedure shown in Fig. 7. The simulation results indicate that the factor of safety of the slope model using the improved gravity increase method is 12. Additional details on the factor of safety can be found in Sect. 7 of this paper.

## 4.2 Initiation and propagation of fractures

When the factor of safety is determined, the critical slip surface is simultaneously obtained. To get an in-depth understanding of the fracture propagation mechanism, the propagation process of fractures and the evolution of force chains during the formation of the critical slip surface are recorded (Fig. 8). The force magnitude is proportional to the thickness of the line in the force chain plots (lower right corner of Fig. 8), in which the blue and green lines denote compressive and tensile forces, respectively. The color of the line segments is obvious in the region where the stress concentration is strong.

Non-homogeneous stresses are distributed throughout the slope owing to the heterogeneity of the slope model. After 2,000 time steps, the compressive stress, which slowly increases from top to bottom under the action of gravity, is initially distributed throughout the slope. The tensile stress exists only in the tips of the fractures (Fig. 8a). Then, the degree of tensile stress concentration increases at the fracture endpoints (Fig. 8b). The fractures (black lines surrounded by a pink circle in Fig. 8) propagate from the tips of the original fractures (red lines) where the tensile stress is concentrated (Figs. 8a and 8b). After the initiation of fracture propagation, the tensile stress (green markings) at the tips of the original fractures dissipates (lighter green markings). A new tensile stress concentration is found at the tips of the propagated fractures, as shown in the force chain plots in Figs. 8b and 8c. The propagated fractures continuously expand downward to the tips of the neighbouring fractures (or rock surface) accompanied by the concentration and release of tensile stress. The orientation of the yellow arrows in Fig. 8 corresponds to the fracture propagation direction. Finally, a decrease in forces, especially the compressive forces (lighter color) throughout the slope, is observed; furthermore, a through-going surface, i.e., the critical slip surface, is formed when the propagated fractures arrive one after another to the tip of the original fractures (Fig. 8d).

## 5 Potential failure and accumulation process

According to the aforementioned results of stability analysis, the factor of safety is extremely high such that the investigated rock slope is highly stable. Nonetheless, the potential failure and the accumulation process are simulated in the present study to obtain good knowledge of the failure mechanism of fractured rock slopes and provide references for similar slope projects. To maintain coherence of analysis, we use the model in Fig. 6 as an example. Section 4 presents an analysis of the formation

process of the critical slip surface of the gravity-increased model. Subsequently, another round of analysis is performed to determine the failure process after forming the critical slip surface.

Figure 9 presents the displacement field images of the slope model in different time steps. In the displacement field plots, particles are colored according to their relative displacement magnitude. Figure 9a, which corresponds to Fig. 8d, represents the displacement field of particles when the critical slip surface is initially formed. The displacement field image

indicates the occurrence of a small deformation in the failed mass above the critical slip surface; the bedrock remains stable without a distinguished gap with the failed mass. The displacements of the failed mass continuously increase, and the largest displacement of the particles (red particles in Fig. 9b) is nearly 0.2 m, which indicates the aggravation of rock failure (Fig. 9b). After 60000 time steps, the failed mass is slowly fractured into many rock blocks along the non-persistent fractures (Fig. 9c). The displacement and size of these blocks vary from one another. The block fragmentation is apparent near the critical

slip surface because of the newly propagated fractures, whereas most of the rock blocks far from the critical slip surface remain intact (Fig. 9d). Ultimately, the failed mass is completely separated from the bedrock (Fig. 9d).

As soon as the failed mass is detached from the bedrock, a large displacement and movement of blocks occurs. To reflect the actual failure and accumulation process of rock slopes in nature, the PFC2D procedure should be manually interrupted to make corresponding adjustments (when the time steps are 80,000, which correspond to Fig. 9d). First, the

particles in the bedrock are deleted to improve computational efficiency, and the critical slip surface is replaced with a rough rigid wall. The frictional properties of the rigid wall are the same as those of the propagated fractures. Then, the DFN model is removed, and the body force in the model is initialized to avoid splitting caused by the release of stress. Finally, the gravity acceleration is restored to 10.0, which corresponds to natural conditions. Figure 10 presents the computed results. In the figure, the blue sections denote the assemblage of rock particles from a macroscopic level while the red lines represent

the bonds (contacts) between particles. For a convenient description, we numbered several rock blocks from 1 to 6.

In Fig. 10a ($5 \times 10^4$ time steps), except for block 2, the blocks rotate under inertia force and gravity and are separated from one another. The blocks near the slip surface are disintegrated into numerous sub-blocks, and even crushed particles. Rock blocks 1, 3, 4, and 5 rotate counterclockwise by approximately 60°, 30°, 260°, and 40°, respectively; whereas block 6 rotates clockwise by 75° (Fig. 10b) when the time steps are $2 \times 10^5$. Block 6 initially crashes to the ground, and the collision

results in the bond breakage between particle blocks (red segment in Fig. 10c). Block 6 is divided into two sub-blocks and some crushed particles. Blocks 4, 5, and 2 successively crash to the ground one after another but their shapes are kept intact, whereas block 3 is split owing to the collision with block 6, as shown in Figs. 10c and 10d. As soon as all crushed particles and blocks fall to the ground or the critical slip surface, the blocks and particles slide or roll forward as a whole and are accompanied by a fragmentation of the block edges (Fig. 10e). Figure 10f presents the final morphology of the rock model.

The final deposit is composed of relatively intact rock blocks and crushed particles, and the blocks pile up above the crushed particles, presenting an inverse grading phenomenon. We also record the final accumulation distance, which is represented by the farthest reach distance of intact blocks (bonds exist in particles in these blocks). The final accumulation distance of the rock slope model shown in Fig. 6 is 28 m.

## 6 Statistical analysis

The slope analysis in Sects. 4 and 5 are based on a SRM model with one DFN model. However, as mentioned in Sect. 2, numerous DFN models can be generated because of the variability of DFN simulation, and the stability analysis of only one SRM model may lead to erroneous results. In the present study, 100 SRM models on the basis of the different DFN models (generated in Sect. 2), are built to conduct the aforementioned analysis. In particular, the critical slip surfaces and safety factors of these models are calculated using the improved gravity increase method following the calculation procedures in 325 Fig. 7. Then, the method mentioned in Sect. 5 is employed to simulate the potential failure and accumulation process.

The factors of safety and the critical slip surfaces of the 100 slope models based on 100 different DFN models significantly vary. For example, Figure 11 exhibits the critical slip surfaces of the SRM models based on DFN models shown in Fig. 4. According to the results, the locations and shapes of the four critical slip surfaces significantly vary, although they all extend along the non-persistent fractures. The factors of safety of these models are 12, 15, 21, and 17.

Figure 12 illustrates the critical slip surfaces of the 100 SRM models on the basis of 100 different DFN models. Fig. 13a present the factors of safety of the 100 models, ranging from 12 to 38. Therefore, the variability of the simulation results should be emphasized in the stability evaluation of fractured rock slopes. The outcome that considers numerous model calculations may lead to a rational result. In the present study, a statistical analysis method is applied to solve this problem. The final critical slip surface and factor of safety are attributed to conservative considerations. The potential critical slip 335 surface constitutes a large failed mass (Fig. 12). The final critical slip surface covers over 90 % of the failed mass to guarantee safety in the rockslide analysis and support. Additional information on the critical slip surface can be found in the research of Zhang et al. (2017). Critical slip surfaces are supposed to have an arc-shaped geometry that differs from their actual linear or broken line morphologies. Arc morphology is easily defined and is convenient for practical engineering designs. $F_s$ is the final factor of safety of the rock slope when the $F$ values of the other 90 models are greater than $F_s$. The 340 result shows that the final factor of safety of the rock slope is 19.

Similarly, the results of the potential failure and accumulation process vary. Figure 14 presents the final accumulation states of the models shown in Fig. 11. The plots indicate that the sizes and quantities of the fractured blocks, as well as the rotating degrees, are significantly different. The final accumulation distances of the models are 28, 34, 35, and 40 m.

Finally, the different accumulation distances, with the minimum value of 15 m and the maximum value of 96 m, of the 345 100 SRM models based on 100 different DFN models are obtained (Fig. 13b). The potential failure and accumulation process based on one SRM model may obtain erroneous results. A statistical analysis method is also applied to solve this problem. To guarantee safety in the rock slope analysis, we attribute the final accumulation distance to conservative considerations. In particular, when the distance values of the other 90 models are lower than a certain value, then the value is the final accumulation distance ($D_a$) of the rockslide. Therefore, the final accumulation distance is 87 m.

## 7 Discussion

Slope failure is a 3D stability problem. Thus, constructing a 3D SRM model for stability evaluation and failure process analysis is highly convincing and accurate. A 3D SRM model also combines rock masses in the BPM with DFN model. However, the quantity of rock particles in a 3D slope is extremely large to conduct an effective calculation in PFC. In addition, the factor of safety obtained by 3D analysis is generally higher than that in 2D analysis. Given that many theories and technologies cannot be established and that their adoption cannot perfectly reflect rock masses at present, the accuracy of analyses may not satisfy engineering project requirements. Moreover, deriving a high factor of safety is sometimes unfavourable. Accordingly, 2D analysis, which is simple and commonly used in reality, is adopted in the present study for the stability evaluation and potential failure analysis of the investigated slope. The 2D analytical result can be regarded a good reference, and it provides a theoretical and practical basis for future initiatives that utilize 3D analysis.

The factor of safety of the investigated slope is extremely high but reasonable. In the field investigation, weak interlayer and through-going discontinuities are not observed. The non-persistent fractures are very developed, which therefore play a vitally important role in the stability of the investigated slope. The strength of the rock bridges (intact rock) is considerably higher than that of the fractures. Therefore, the obtained factor of safety is extremely high and is thus reasonable. In addition, the effects of water (rainfall) and earthquakes were ignored in the present study. However, the accuracy of the calculation result would increase if rainfall (seepage analysis) and earthquakes (kinetic analysis) are considered. This topic will be the direction of our future research.

The failure process is unlikely to occur in the investigated rock slope unless it is subjected to significant environmental changes, such as earthquakes, rainfall, unloading, or overloading. Nonetheless, the potential failure process is simulated in this study to understand the rockslide mechanism and subsequently provide a good reference for similar slope projects. For example, the size and motion of rock blocks can be utilized to predict risk degree, and the final accumulation of deposits can contribute to hazardous area division. The final arrangement of deposits (a combination of blocks and crushed particles) provides a good explanation for the inverse grading of rock avalanche reported by Cruden and Hungr, (1986), Imre et al. (2010), and Wang et al. (2012).

DFN simulation presents a high degree of variability and may provide erroneous results. A statistical analysis based on numerous SRM models with different DFN models similar to those performed in the present study can reduce errors and provide conservative results for slope support. Finally, although the final results of the factor of safety, critical slip surface, and accumulation distance can guarantee safety in the rockslide analysis and support, they are not the exact values. Statistical analysis provides a new method for deriving an in-depth understanding of solid earth, where specific locations and characteristics of geological materials and structures remain unknown, such as discontinuities, especially for small-scale non-persistent fractures. Meanwhile, new theories and technologies are required to obtain precise forecasts with respect to the range values characterized by statistical methods.

**8 conclusion**

The present study combines several methods, namely, DFN simulation, SRM approach, and statistical analysis, to conduct stability evaluation and potential failure process analysis of a fractured rock slope in Laohuding Quarry in Jixian County,
Tianjin. The SRM technique is utilized to generate a slope model with non-persistent fractures in the form of a DFN. The factor of safety is determined on the basis of the improved gravity increase method. The formation of a critical slip surface is also investigated. The potential failure and accumulation processes are simulated to provide a reference for similar slope projects. Numerous slope models are calculated, and the final results of the safety factor, critical slip surface, and accumulation distance are determined by statistical analysis. The major findings are summarized as follows.

(1) The slope model with non-persistent fractures can be effectively constructed on the basis of SRM technology. The instability of the slope model can be attained by combining the improved gravity increase method or the strength reduction method, thereby obtaining the safety factor and critical slip surface. An innovative formula to calculate the safety factor is proposed by considering stress concentration and the calculation principle of PFC2D.

(2) Fracture propagation is closely related to stress concentration. Fractures initially propagate from the tips of the
original fractures where the tensile stress is concentrated. Then, the stress is released, and a new stress concentration occurs at the tip of the propagated fractures when the fracture propagates downward to the neighbouring fractures. The critical slip surface is formed by the coalescence of preexisting fractures and newly propagated fractures.

(3) In the initiation of failure, the failed mass is fractured into rock blocks along the preexisting fractures. Then, most blocks rotate and collapse under inertia force and gravity. Several blocks are split into sub-blocks owing to the collision
between the blocks and the ground. The final deposit is composed of intact blocks and crushed particles, presenting inverse grading phenomena.

(4) The critical slip surfaces, factors of safety, and accumulation distances of the slope models with different DFN models vary. Therefore, the final outcome is obtained by statistical analysis. It ensures engineering safety for rockslide analysis and support. The factor of safety (reserve) of the studied rock slope is determined to be 19. The critical slip surface
is confirmed, as shown in Fig. 12. The final accumulation distance is 87 m.

**Data availability.** The data from the research findings cannot be shared now, because these data are also part of the future research.

**Author contributions.** WZ initiated this research, analysed the simulation results, and revised the manuscript. JW wrote software code and wrote this paper. PX, JL and BS conducted field investigation and recorded field data. FW and CC dealt
with the field data. XC and JQ guided and collated the software code.

**Competing interests.** The authors declare that they have no conflict of interest.

**Acknowledgements.** This work was supported by the National Natural Science Foundation of China (grant nos. 41877220 and 41472243), the National Natural Key Science Program Foundation (grant no. 41330636), the National Key Research and

Development Plan (grant no. 2017YFC1501000), and the Science and Technology Project Plan of Northeast Electric Power
Company (grant no. K2018N-19).

**Financial support.** This research has been supported by the National Natural Science Foundation of China (grant nos. 41877220 and 41472243), the National Natural Key Science Program Foundation (grant no. 41330636), the National Key Research and Development Plan (grant no. 2017YFC1501000), and the Science and Technology Project Plan of Northeast Electric Power Company (grant no. K2018N-19).

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

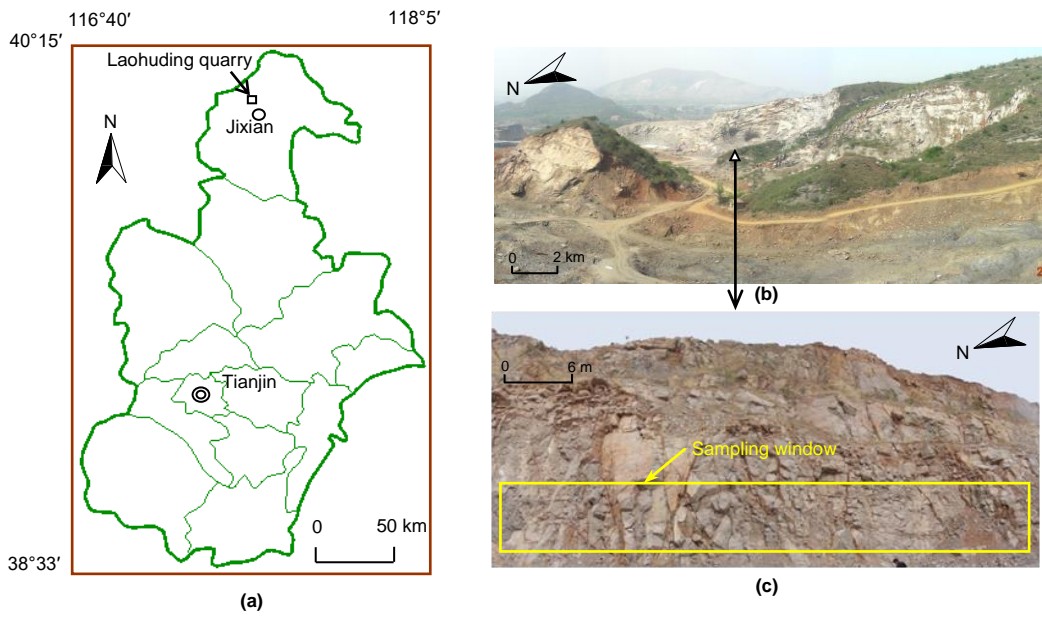


**Figure 1:** Location of the Laohuding Quarry and investigated rock slope. **(a)** location of Laohuding Quarry in Tianjin City, China; **(b)** image of Laohuding Quarry and the location of the investigated rock slope; **(c)** image of the investigated rock slope with a strike of 200 °.

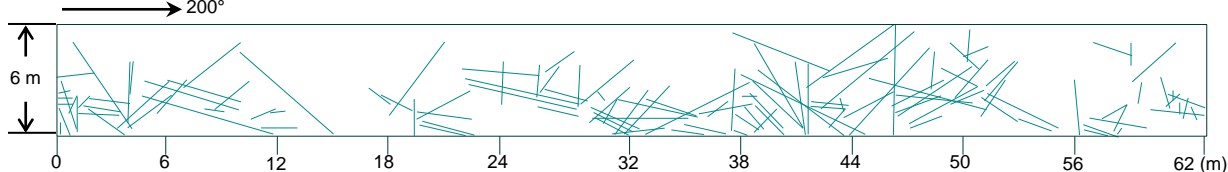

**Figure 2:** 2D trace chart of collected fractures.

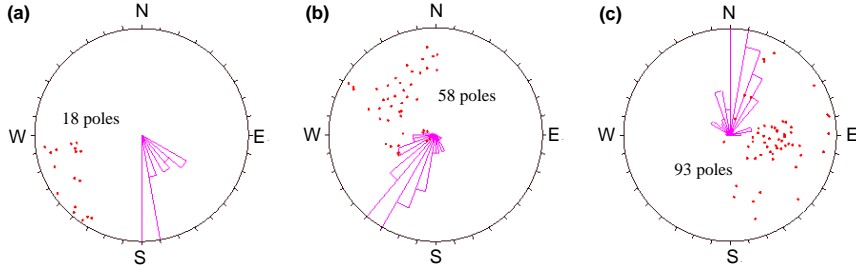


**Figure 3:** Poles and strike rose diagrams of the fracture sets. **(a)**–**(c)** are fracture sets 1, 2, and 3.

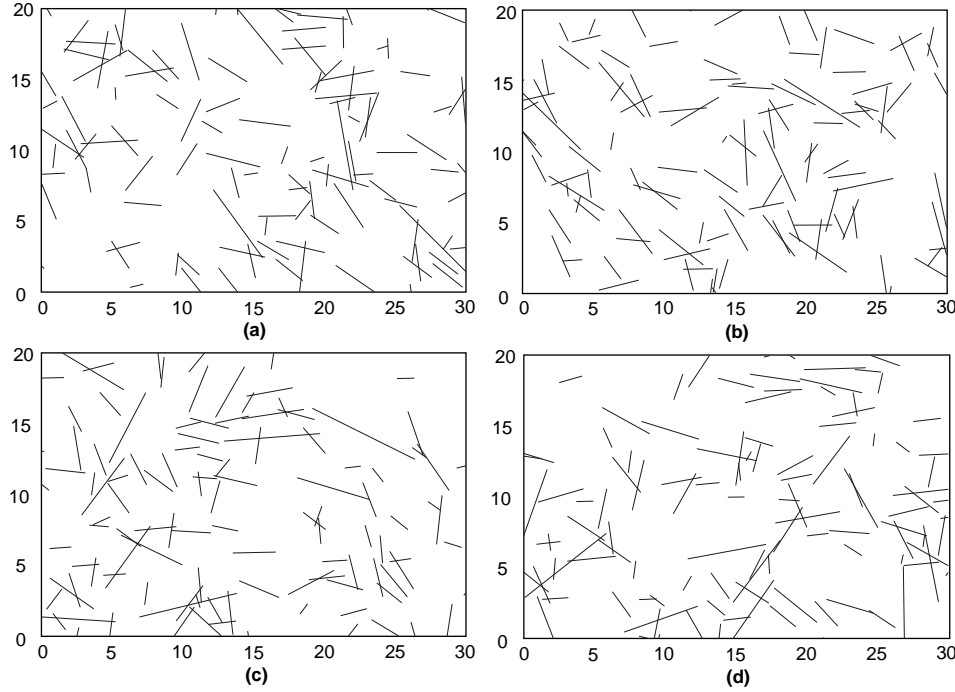

**Figure 4:** Variability of DFN simulation. **(a)**–**(d)** are the DFN models in four simulations. The line segments in the rectangle frame represent the fractures, and the left boundary of the frame represents the exposed surface.

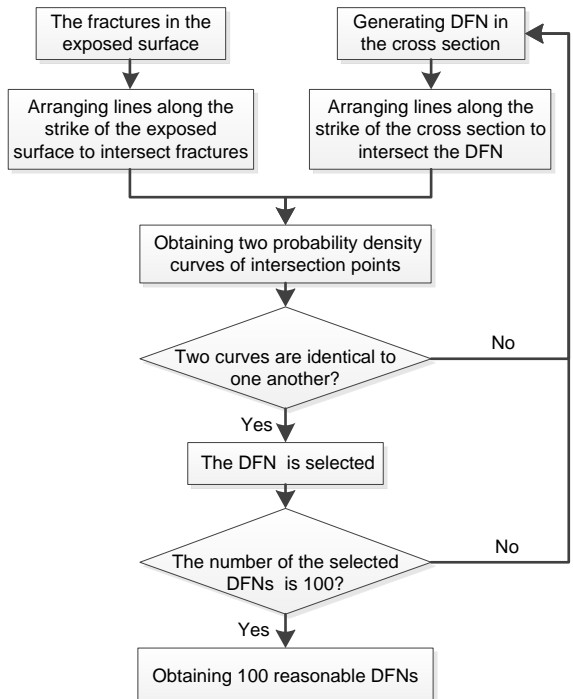


**Figure 5**: The procedure of selecting reasonable DFNs.

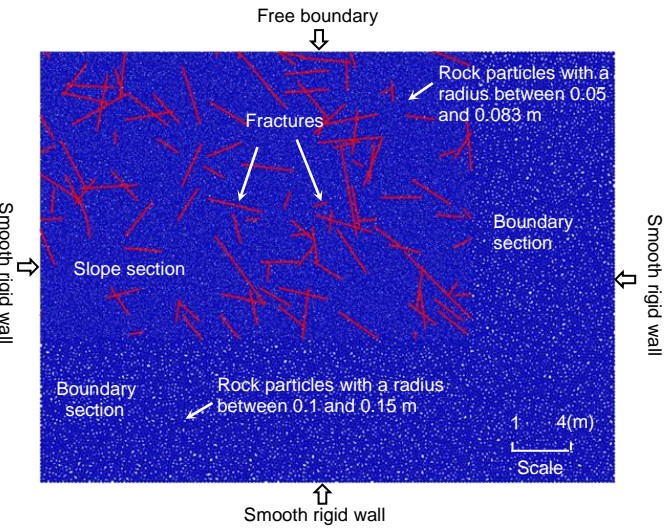

**Figure 6:** SRM model of the investigated fractured rock slope. The left boundary is the exposed surface.

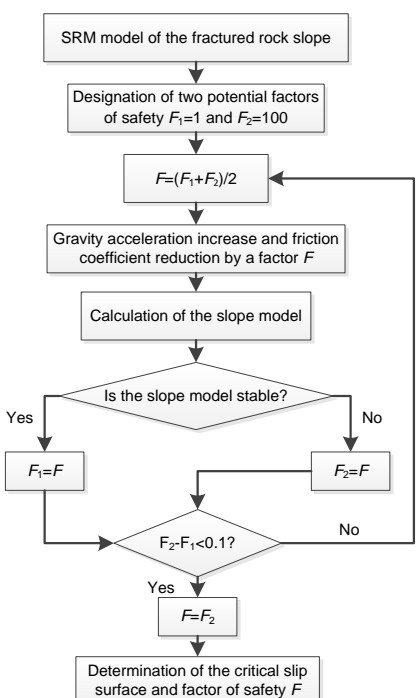

**Figure 7:** Program flow chart for the slope stability analysis.

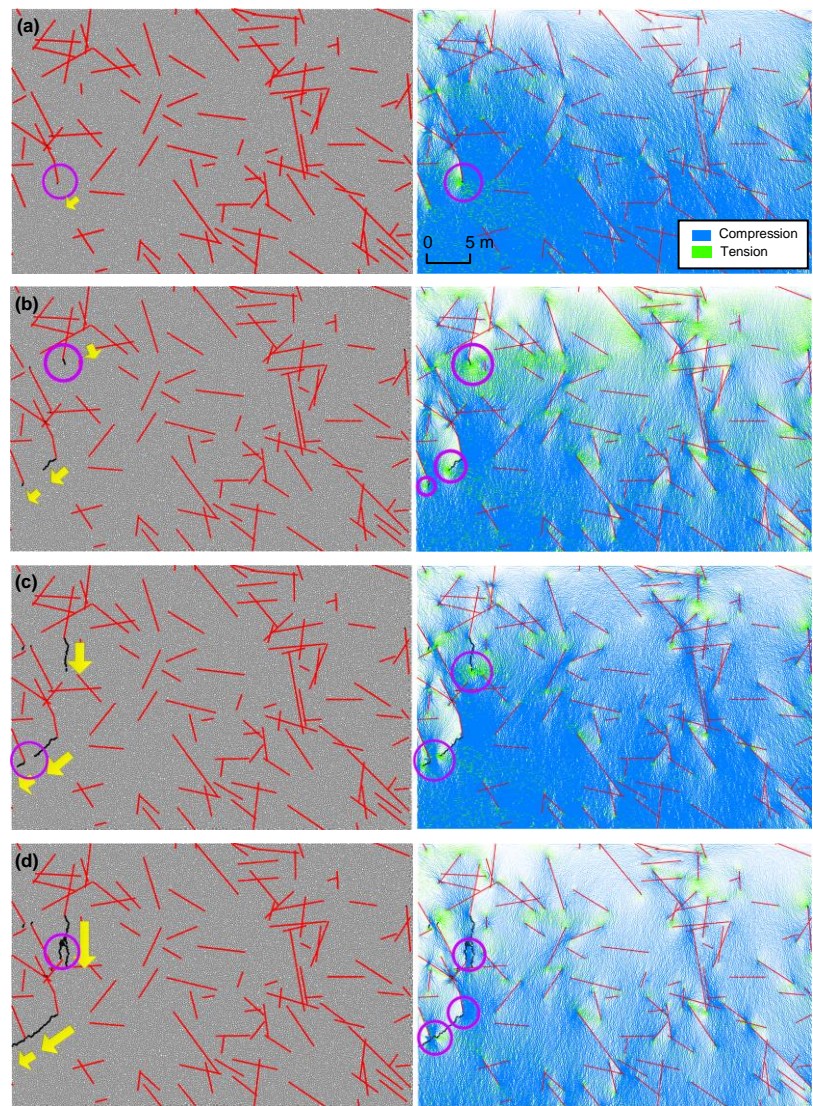

**Figure 8:** Formation process of the critical slip surface (drawings on the left) and force chain plots (drawings on the right). The time steps of **(a)**–**(d)** are $2\times10^3$, $5\times10^3$, $10^4$, and $2\times10^4$.

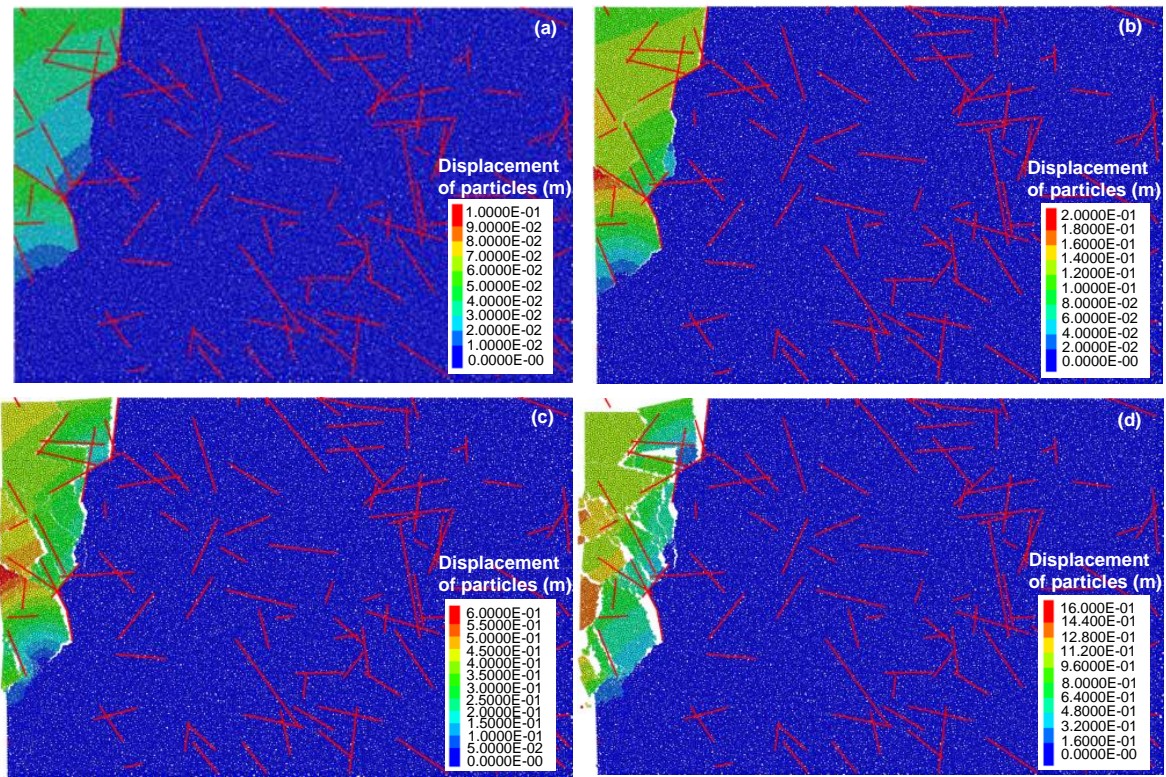

**Figure 9:** Particle displacement field of the fractured rock slope. The time steps of **(a)**–**(d)** are $2\times10^4$, $4\times10^4$, $6\times10^4$, and $8\times10^4$.

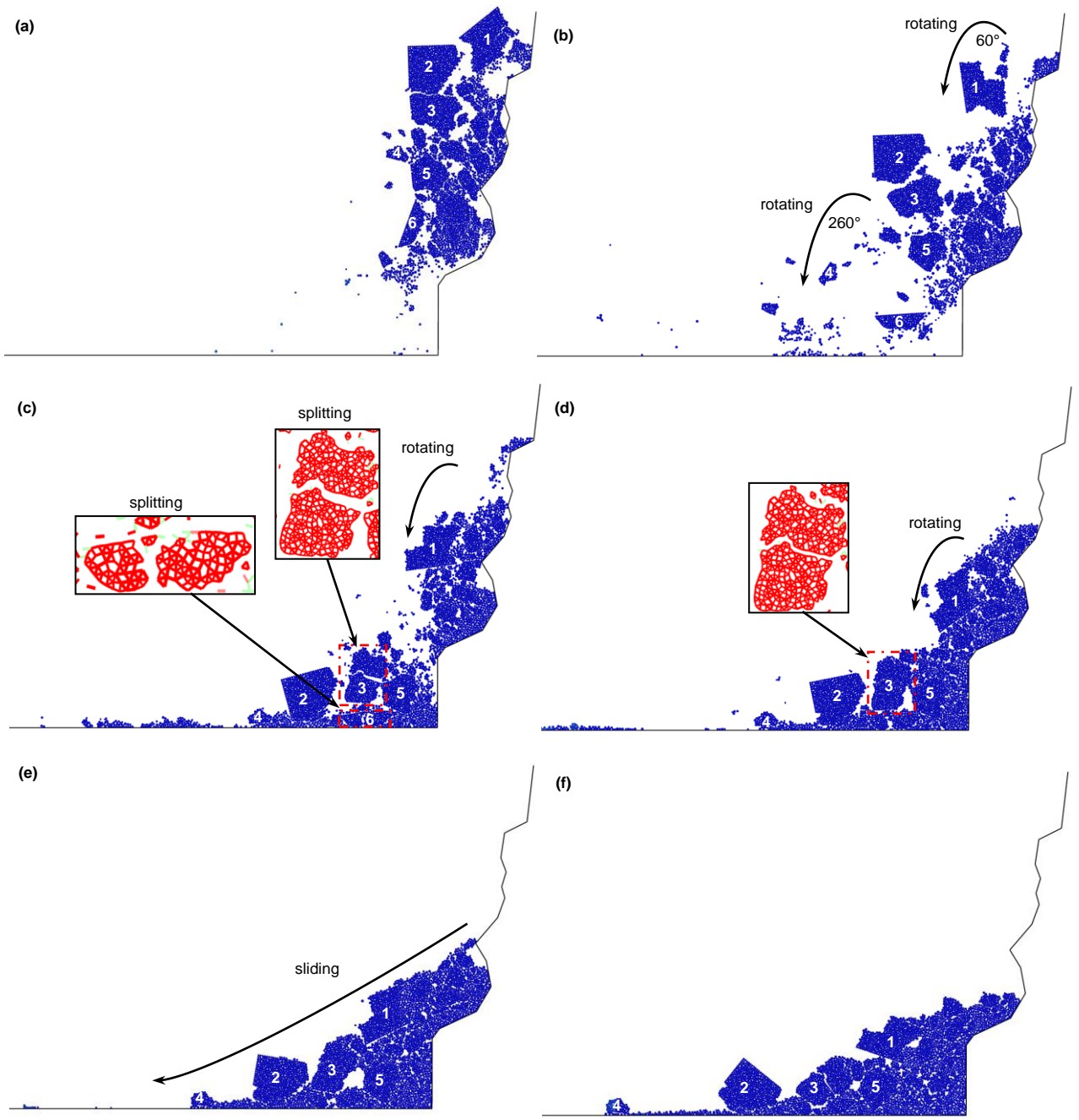

**Figure 10:** Failure and accumulation process of the rock slope. The time steps of (**a**)–(**f**) are $5 \times 10^4$, $2 \times 10^5$, $4 \times 10^5$, $8 \times 10^5$, $1.6 \times 10^6$, and $3.2 \times 10^6$.

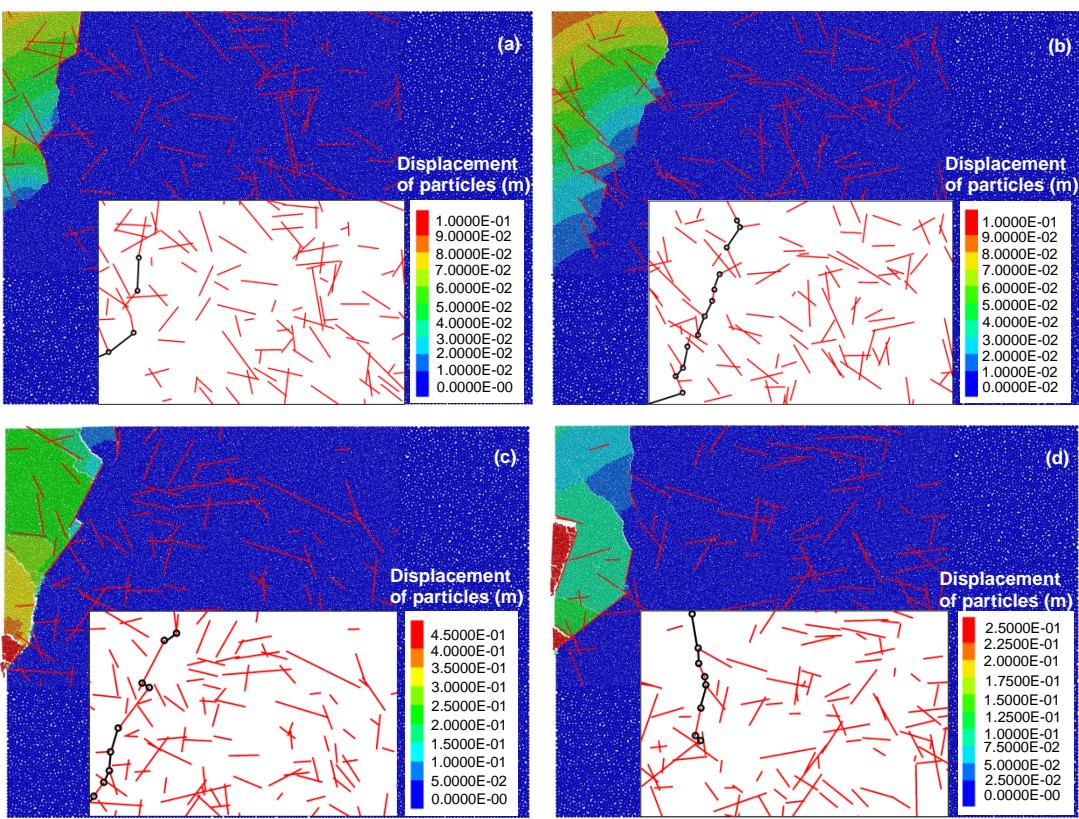

**Figure 11:** Four different critical slip surfaces of rock slopes with four different DFN models. The factors of safety of **(a)**–**(d)** are 12, 15, 21, and 17.

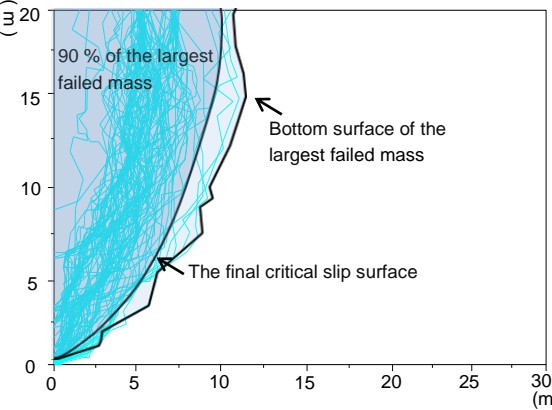

**Figure 12:** Potential critical slip surfaces of 100 models and the final critical slip surface of the investigated rock slope.

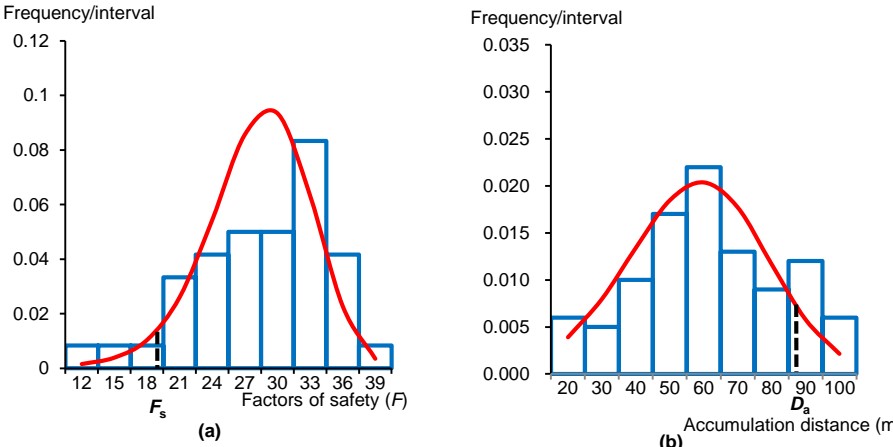

Figure 13: Statistical results of 100 slope models. (a) the final factor of safety $F_s$ of the investigated rock slope; (b) the final accumulation distance $D_a$ of the investigated rock slope.

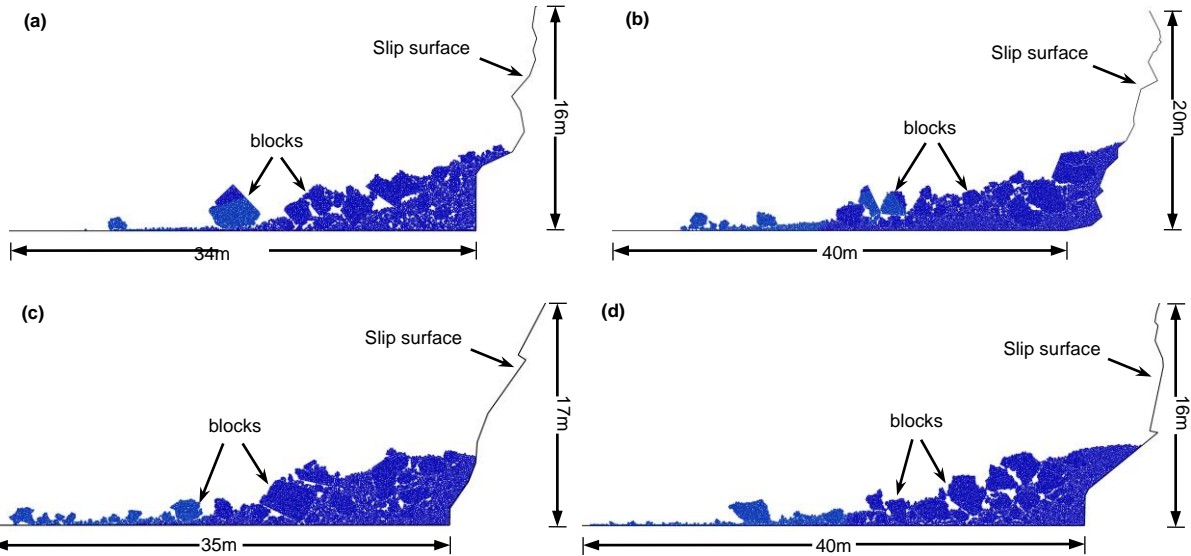

Figure 14: Ultimate accumulation states of rock slopes with different DFN models (partly). The final accumulation distances of (a)–(d) are 28, 34, 35, and 40m.

Table 1: Parameters for discrete fracture network (DFN) simulation.

| Fracture set | Fracture number | Dip Direction (°) | Dip Angle (°) | Trace length surveyed (m) | Corrected (m) | Std. (m) | Distribution type | $R_0$ | $R_1$ | $R_2$ | $P_{21}$(m/m$^2$) | $K$ |
|---|---|---|---|---|---|---|---|---|---|---|---|---|
| 1 | 18 | 39.5 | 87.3 | 2.16 | 2.42 | 1.2 | Gamma | 0.89 | 0.11 | 0 | 0.11 | 17.1 |
| 2 | 58 | 307.4 | 44.7 | 2.71 | 2.84 | 1.7 | Log-normal | 0.83 | 0.14 | 0.03 | 0.45 | 10.3 |
| 3 | 93 | 110.2 | 31.7 | 2.33 | 2.49 | 1.2 | Gamma | 0.87 | 0.13 | 0 | 0.63 | 9.1 |

**Table 2:** Values of the BPM parameters.

| Particle parameters | | Parallel bond parameters | |
|---|---|---|---|
| Particle density (kg/m³) | 2650 | $E_b$ (GPa) | 23 |
| Minimum particle radius (m) | 0.05 | $k_{nb}/k_{sb}$ | 1.25 |
| Radius ratio | 1.66 | Normal strength (MPa) | 6.25 |
| Friction coefficient | 0.35 | Internal friction angle (°) | 38 |
| $k_n/k_s$ | 1.25 | Cohesion (MPa) | 6.25 |
| $E_c$ (GPa) | 23 | Radius multiplier | 1 |

**Table 3:** Comparison of macro-properties determined by numerical and laboratory tests.

| Macro-parameters | E (Gpa) | Poisson's ratio | Friction angle (°) | Cohesion (Mpa) | $\sigma_c$ (Mpa) |
|---|---|---|---|---|---|
| Numerical tests | 35.7 | 0.23 | 22 | 6.22 | 37.8 |
| Laboratory tests | 35 | 0.24 | 21.58 | 6.125 | 38.3 |

**Table 4:** Calibrated smooth-joint parameters and results of numerical and laboratory tests

| Smooth-joint parameters | | Test results | | |
|---|---|---|---|---|
| | | Parameters | Numerical tests | Laboratory tests |
| Normal stiffness $\overline{k}_{nj}$ (GPa/m) | 10 | Normal stiffness (GPa/m) | 7.042 | 7 |
| Shear stiffness $\overline{k}_{sj}$ (GPa/m) | 8 | Shear stiffness (GPa/m) | 3.415 | 3.4 |
| Coefficient of friction $\mu_j$ | 0.466 | Friction angle (°) | 17.87 | 18 |