# Peer review of "Stability evaluation and potential failure process of rock slopes characterized by non-persistent fractures"

_Natural Hazards and Earth System Sciences, 2020_

## Referee Comment (RC1) · Anonymous Referee #1 · 16 May 2020

The manuscript presents a comprehensive approach that combines several well-established methods to carry out both stability and runout analyses of a fractured rock slope in Laohuding Quarry (Jixian County, China). Specifically, the discrete fracture network modelling technique is performed to generate non-persistent discontinuities according to the fractures collected in the field. Subsequently, synthetic rock mass approach is applied to simulate the slope model using 2D particle flow code software, and a stability analysis is carried out based on the improved gravity increase method. Finally, by analysing 100 slope models generated with different DFN models, the critical slip surface, factor of safety, and accumulation distance are discussed through a statistical analysis.

GENERAL COMMENT As a general comment, the paper is clear, well structured, and accurately describes tools and techniques that are of great interest to those involved in landslide practice. The results of the proposed methodology seem quite promising, but, in my opinion, the weakest point is related to the selected case study, whose slopes seem to be too stable and, therefore, real collapses are not presented nor discussed. These could have been fundamental for reliable validation of the performed analyses, regarding both the triggering and runout phase of the rockslide. My recommendation is therefore: accepted with minor revisions, taking into consideration the following specific comments SPECIFIC COMMENTS L 69-70: Authors state: "These slopes may become hidden dangers (e.g., geological disasters) and pose potential threats to people and nearby equipment". This sentence is in strong contrast with the results from the stability analyses, since the safety factors range from 25 to 75. I'm wondering if the investigated slope is really too stable or the safety factors could have been somehow overestimated. Nevertheless, no event seems to have occurred in recent times from the investigated slope; this is a pity, since real events could have been extremely useful for validating the performed analyses. Why did Authors choose to test the proposed procedure to this slope? L 92-93 and figure 3: the division of the detected discontinuities into sets is rather questionable, since data dispersion is indeed too high. Yet, this should not affect model reconstruction too much, as dispersion is also considered in the generation of artificial discontinuities within the DFN model. L184: I suggest not to take for granted what a "fish function" is; please explain for those unfamiliar with numerical modelling L255-259: The modelling results are nice and look quite reasonable, but without validation they seem to be an end into themselves. Did any real event occur from the investigated slope in recent times? Are there any detached blocks to compare their volume and shapes with the simulated ones? What about their runout? In my opinion the reliability of the proposed procedure needs to be demonstrated. L260: Please change section 6 title, since "stability analysis" has already been used for section 4. Maybe, "Statistical analysis"? L281: please change are 43.5 to is 43.5 L327 and 334: To trigger instability, Authors used the gravity increase method proposed by Meng

et al. (2015). It is not clear to me if Authors made some amendments to this method, or if they used it as is. If so, in my opinion, the method cannot be defined as "innovative". Figure 1: Fig 1b: image is not clear; I suggest deleting the text superimposed over the image. Fig 1c: if the image contains the window reported in Figure 2, please add the box limits. Figure 7 In my opinion, the picture in picture representation is misleading. If possible, add all the drawings in a single image, otherwise, split in two different figures. Figure 8: to improve image resolution, I suggest cropping the images to the fractured sectors. Figure 9: sometimes the timestep count is represented in decimal notation and sometimes in scientific notation, please uniform.

---

## Referee Comment (RC2) · Anonymous Referee #2 · 25 May 2020

The paper describes the application of the synthetic rock mass (SRM) approach for the investigation of a rock slope within a limestone quarry. The authors briefly summarize the input data used for the creation of discrete fracture networks (DFNs) which are then implemented into a bonded particle code (i.e. PFC2D) in order to investigate the potential failure mechanism of the slope. The numerical analyses are undertaken using a modified gravity increase method, and high factors of safety (>25) are computed for all the DFN realizations. The study represents a starting point for future studies focusing on the analysis of higher slopes, also in order to evaluate and simulate potential support methods.

[Figure]

The paper is well written, easy to read (except a few sentences throughout the paper requiring some rephrasing), and overall, well structured.

I have a few general comments about the manuscript. Then, I will also provide line-to-line observations/comments/feedback/criticism as needed.

The first general comment is that the investigated slope is quite low (20 m in the simulated models). It is true that in certain conditions brittle damage can develop even in low stress conditions, especially when tensile strength is exceeded or in case of significant stress concentrations. However, in this case it seems that the amount of rock bridges along the critical path(s), would most likely stabilize the slope, except perhaps for small blocks at surface. It is very unlikely that the real slopes will ever fail with the simulated mechanism, unless smaller fractures are included in both the data collection and DFN. This is clearly confirmed by the high factors of safety computed in the simulations.

This brings me to the second, important comment: the model input data, which seems the "weak point" of this manuscript. It is not clear what technique was used for the initial data collection, but the DFN used in the models seems to rely on and include only the larger discontinuities. This results obvious by visually comparing figure 1c and figure 2 (which seems to depict a much more fractured rock mass). Considering the input DFN, the high factors of safety computed makes sense. However, the question is: "is this DFN a realistic representation of the real rock mass?". The issue with the input data may have significantly impacted the numerical results, in terms of factor of safety, and location of the critical path. A "work around" would be to decrease the simulated strength of the intact material, to account for the smaller fractures that are not implemented as discrete discontinuities in the DFN.

In conclusion, in its present form, the manuscript seems quite conceptual, in that there is a seemingly weak connection between the model and the actual rock mass. While the simulations may indeed reproduce a realistic fracture propagation mechanism, the actual process is very unlikely to occur in the simulated slope, or any slope with similar height, lithology, and structural configuration. In view of this, this paper should be significantly improved (major revisions) with regard to the model input data and their description. Either by a) including the fractures that seem to be missing in the DFN, or b) demonstrating in a clearer manner the similarity between the DFN and the rock mass. This could include better pictures, with close-ups and, importantly, scales. Showing the mapped traces onto the photo of the slope could also greatly improve the clarity.

In the following, I will provide line-by-line comments. Line 40: authors should refer to "the non-persistent fractures in these works", rather than "these non-persistent fractures". Line 51: SRM has been used also for underground applications, including mining and hydraulic fracturing. Lines 51-53: perhaps these two sentences can be merged. However, the first sentence requires rephrasing, as it seems some words are missing. Line 58: authors should state the country the investigated site is located in. Line 71-72: this sentence can be improved. Perhaps the slopes are higher to the south, rather than the quarry area itself. Also, from this sentence it is not clear whether the "mountain "is a ridge oriented north-south, or the if the bedding are dipping to north (or south). Line 79: is the formation thick, or the limestone layers? Either way, how thick? Karst phenomena are not obvious, meaning there is not any, or that they are not or scarcely visible? Lines 80-83: the last two sentences could be merged. The term "intermittent" for discontinuities is somewhat inaccurate (or simply very rarely used, to my knowledge) – perhaps simply "non-persistent" is more appropriate. Additionally, one would expect that bedding would be very persistent. Will this play a role? Although it is true that observing the bedding trace does not necessarily imply a fully persistent plane with no tensile strength. Line 86-92: A 62 by 6 m is a large area to perform systematic discontinuity mapping (i.e. using traditional field techniques or short range remote sensing methods), and 169 discontinuities seems a low figure – what is the cut-off limit you considered (i.e. the smallest fracture that was considered). Looking at figure 2, it seems that the location of the mapped discontinuities is slightly biased towards the bottom of the window. Because of this, I would assume that the mapping was performed using traditional field methods, rather than remote sensing techniques. Either way, this

should be mentioned, also to address or acknowledge the potential limitations of the methodology used, in terms, for instance, of orientation bias. Speaking of orientation bias, the fracture set 1 is suggested to be less represented – this might be due to its orientation very similar to the slope, while sets 2 and 3 are almost perpendicular. It is nt clear whether this was kept into account. Line 90: Reference here seems out of place, unless the result came from that specific work. Line 94-95: This sentence requires rephrasing. I suggest starting from the issue of the trace length bias, and then stating the reason, rather than the contrary. Line 99: I recommend "and 'investigate the' potential failure mechanism". Lines 105-107: It seems the authors suggest that the fracture intensity in a section is a function of the orientation of the set, with respect to the north. This is a bit counter-intuitive, as the orientation of the rock face (and specifically the angle with the fracture set) is surely more relevant than the azimuth (angle with the North) of the fracture – which in fact should not be that important. More detail on the method should be provided to improve clarity. Line 106: P21 is "fracture intensity", not "fracture density" (which is P20 in the 2D case). Line 117-118: The second statement seems to suggest that different input data ("fracture characteristics") were used for the four DFN realizations. Lines 120-123: Perhaps a sketch would help the reader understanding the procedure. Also, I believe this procedure is performed in 2D. If so, I suggest to use slope "profile" instead of "surface" – this would make the procedure easier to understand for the reader (especially if no figure is provided). Line 130: even more importantly, SRM is used to simulate the brittle propagation of fractures, and thus the brittle behavior of rock masses. Line 134: it would be good to provide a couple of examples (even in brackets) of the micro-properties that are used as input. Line 160: perhaps "interpenetrate" or "compenetrate" is a better term than "pass through". Lines 186-188: this assumption is perhaps more adequate considering the low stress conditions that characterize the real slope. Line 190: Just a comment here. As the authors know, this approach may be "risky" in other conditions (i.e., high stress/high slopes) as it may cause a "shock" in the model, causing excessive damage in the slope, compared to a progressive excavation (or a progressive removal of the boundary), which generally

is more representative of a "real world" situation. Lines 194-197: It is unclear what are the benefits of decreasing the friction coefficient while increasing gravity. Intuitively, the gravity increase already would induce an increase in shear stresses compared to the initial state (very much like increasing the density). How is this double effect (increase in shear stress, decrease in shear strength) accounted for in the FoS calculation? And why just the friction, and not the cohesion? The paper would benefit from a more detailed explanation of the method employed. Line 202: this seems a very stable slope. Expectedly, in view of the amount of rock bridges along the rupture surface, which may be estimated at about 30-40%, according to figure 7. Line 232 (and after): Perhaps it will be better to refer to the "slip mass" as "failed", or "detached" mass/volume/material. Line 255: I recommend referring to the "model geometry" or "morphology", rather than "shape". Line 272: A variability 25-75 is indeed very high. Perhaps this variability be lower if a more realistic DFN (i.e. inclusive of smaller fractures) was to be simulated. Absolute values would be lower, for sure. Line 290: again, just a comment. The limitations of this estimation is that is assumes that the base of the model is constituted by strong rock, likely with high coefficient of restitution, and the distribution of the failed mass over this distance is not considered. Lines 301-303: this sentence is unclear and requires rephrasing. Lines 303-305: I agree with the authors here: rock bridges are multiple orders of magnitude stronger than discontinuities, and this justifies the high FoS. The questions, however, is: are these estimations accurate and representative of the real situation? Figure 4 shows a rock mass significantly more fractured that the DFNs employed in this study, where the slope is formed by very large, intact blocks.

Comments on figures/tables Figure 1c: a scale and possibly a north arrow is required Figure 7: a legend bar (stress) and scales are needed for clarity Figures 8 and 10: the use of an uniform color bar and legend would enhance the comparison of the states depicted by each sub-figure. Table 4: I recommend using the same order for micro, numerical, and lab parameters: friction, normal, and tangential stiffness.

2020-58, 2020.

---

## Author Comment (AC2) · 2 Jul 2020

Dear referee:

Thank you very much for your valuable comments and suggestions. These comments are all valuable and helpful for revising and improving our paper, as well as the important guiding significance to our researches. The main corrections and the responses are listed as follows.

Responses to general comments

Comment 1: The first general comment is that the investigated slope is quite low (20 m

[Figure]

Interactive
comment

in the simulated models). It is true that in certain conditions brittle damage can develop even in low stress conditions, especially when tensile strength is exceeded or in case of significant stress concentrations. However, in this case it seems that the amount of rock bridges along the critical path(s), would most likely stabilize the slope, except perhaps for small blocks at surface. It is very unlikely that the real slopes will ever fail with the simulated mechanism, unless smaller fractures are included in both the data collection and DFN. This is clearly confirmed by the high factors of safety computed in the simulations.

This brings me to the second, important comment: the model input data, which seems the "weak point" of this manuscript. It is not clear what technique was used for the initial data collection, but the DFN used in the models seems to rely on and include only the larger discontinuities. This results obvious by visually comparing figure 1c and figure 2 (which seems to depict a much more fractured rock mass). Considering the input DFN, the high factors of safety computed makes sense. However, the question is: "is this DFN a realistic representation of the real rock mass?". The issue with the input data may have significantly impacted the numerical results, in terms of factor of safety, and location of the critical path. A "work around" would be to decrease the simulated strength of the intact material, to account for the smaller fractures that are not implemented as discrete discontinuities in the DFN.

In conclusion, in its present form, the manuscript seems quite conceptual, in that there is a seemingly weak connection between the model and the actual rock mass. While the simulations may indeed reproduce a realistic fracture propagation mechanism, the actual process is very unlikely to occur in the simulated slope, or any slope with similar height, lithology, and structural configuration. In view of this, this paper should be significantly improved (major revisions) with regard to the model input data and their description. Either by a) including the fractures that seem to be missing in the DFN, or b) demonstrating in a clearer manner the similarity between the DFN and the rock mass. This could include better pictures, with close-ups and, importantly, scales. Showing the

mapped traces onto the photo of the slope could also greatly improve the clarity.

Response: Thank you very much for your comment. It is really true that the investigated slope can hardly fail considering the low height, a large amount of rock bridges along the slip surface. This is also confirmed by the high factors of safety. When it is subjected to significant environmental changes, such as earthquakes, rainfall, unloading, or overloading, the failure may occur following the potential failure mechanism.

The sampling window method (Kulatilake and Wu 1984) is used to collect fracture data in the present study. It is really true that the generated DFNs rely on and include the larger fractures since only fractures with the length larger than 1.5 m were measured in the field. The amount of fracture with the length smaller than 1.5 m are extremely large, which goes beyond the artificial measurement. Considering that small fractures have a little effect on slope stability, we take the cut-off limit of 1.5 m.

On the basis of the collected fracture data in the exposed rock surface, the DFN is a most possible representation of the real rock mass. However, it is really true that the DFN is more accurate and the numerical results are more reasonable if smaller fractures are taken into consideration. In the revised manuscript, although there is no way to recollect fractures with the length smaller than 1.5 m, we reduced the strength of the intact material to remedy the lack of smaller fractures in DFN. Specifically, particle parameters which influence the strength of intact material, including the friction coefficient, tensile strength, and cohesion of particles are synchronously reduced by the same reduction factor (Bonilla-Sierra et al., 2015; Sun et al., 2014).

The new modelling results indicated that the factors of safety are indeed lower with the decrease of strength parameters, indicating the lack of smaller fractures indeed makes the factors of safety high. Nevertheless, the location of the critical slip surface remains consistent regardless of the strength parameter since the slip surface is always composed of pre-existing fractures and new-propagated ones. Similarly, the potential failure process is roughly identical to that of previous simulation since both

are conducted in the natural condition (i.e., the gravity acceleration is 10.0).

In the revised manuscript, corresponding content of parameter reduction and the new modelling results are rewrote, which is modified a lot and thus not described in the responses. In addition, the scales were added in all required figures and the location of the mapping window has been added in Fig. 1c. We also carefully checked the whole manuscript and considered all of your line-by-line comments. Our responses to the line-by-line comments are as follows.

Responses to line-by-line comments

Comment 2: Line 40: authors should refer to "the non-persistent fractures in these works", rather than "these non-persistent fractures".

Response: Thank you very much for your correction. In the revised manuscript, we have changed "these non-persistent fractures" to "the non-persistent fractures in these works".

Comment 3: Line 51: SRM has been used also for underground applications, including mining and hydraulic fracturing.

Response: Thank you very much for your comment. We carefully searched the literatures regarding underground applications of SRM approach. It is really true that SRM approach has been widely used in mining and hydraulic fracturing. Therefore, we added corresponding description as "SRM models have been primarily used to simulate failure and deformation of fractured rock slopes (Bonilla-Sierra et al., 2015; Elmo et al., 2013), simulate hydraulic fracturing in naturally fractured reservoirs (Damjanac and Cundall, 2016), and estimate rock mass strength, fragmentation and micro seismicity in caving mines (Lorig et al., 2017)".

Comment 4: Lines 51-53: perhaps these two sentences can be merged. However, the first sentence requires rephrasing, as it seems some words are missing.

Response: Thank you very much for your suggestion. The latter sentence is the explanation of the former one; thus, these two sentences can indeed be merged. In the revised manuscript, we merged the two sentences as "DFN simulation included in SRM modeling program presents a significant variability, which means numerously possible realizations of 2D fracture systems exist given specified input parameters (Pine et al., 2006)."

Comment 5: Line 58: authors should state the country the investigated site is located in.

Response: Thank you very much for your suggestion. In the revised manuscript, we stated the country the investigated site is located in and described as "This study proposes a comprehensive approach that combines several well-established methods to conduct a stability evaluation and failure process analysis of a fractured rock slope in Tianjin City, China."

Comment 6: Line 71-72: this sentence can be improved. Perhaps the slopes are higher to the south, rather than the quarry area itself. Also, from this sentence it is not clear whether the "mountain "is a ridge oriented north-south, or the if the bedding are dipping to north (or south).

Response: We are very sorry for our unclear description. Actually, all sentences from Line 71 to 73 contribute to the description of the study area (i.e., the quarry area). Therefore, it is true that the quarry area is higher in the north. The "mountain" in this place refers to the monoclinal mountains striking south-north. In the revised manuscript, we rewrote these sentences as "The Laohuding Quarry area is characterized by the low-mountain terrain, which is higher in the north than in the south. The highest and lowest altitudes of the quarry area are 160 m and 60 m, with a relative elevation of 100 m. A majority of monoclinal mountains striking south–north exist in this area. The average slopes of the mountains in the east and west of the quarry area are 25° and 30°, respectively (Fig. 1b)."

Comment 7: Line 79: is the formation thick, or the limestone layers? Either way, how

thick? Karst phenomena are not obvious, meaning there is not any, or that they are not or scarcely visible?

Response: We are very sorry for unclear description. Actually, we initially aims to say "The limestone is moderately weathered" rather than "The limestone is moderately thick". In the revised manuscript, we corrected it. "Karst phenomena are not obvious" means that the karst phenomena are scarcely visible in the study area due to low precipitation and the lack of groundwater. In the revised manuscript, we rewrote this sentence as "Karst phenomena are scarcely visible due to low precipitation and groundwater shortage."

Comment 8: Lines 80-83: the last two sentences could be merged. The term "intermittent" for discontinuities is somewhat inaccurate (or simply very rarely used, to my knowledge) – perhaps simply "non-persistent" is more appropriate. Additionally, one would expect that bedding would be very persistent. Will this play a role? Although it is true that observing the bedding trace does not necessarily imply a fully persistent plane with no tensile strength.

Response: Thank you very much for your suggestion. It is really true that the term "intermittent" for discontinuities is rarely used. In the revised manuscript, we used "non-persistent" to substitute it as you suggested. Field observation demonstrated that no bedding planes, faults, folds, and shear zones are developed in the rock exposure, which we further interpreted in the revised manuscript. Therefore, the bedding plays no role in stability analysis. It is non-persistent fractures that play the most significant role in the slope stability and potential failure process.

Comment 9: Line 86-92: A 62 by 6 m is a large area to perform systematic discontinuity mapping (i.e. using traditional field techniques or short range remote sensing methods), and 169 discontinuities seems a low figure – what is the cut-off limit you considered (i.e. the smallest fracture that was considered). Looking at figure 2, it seems that the location of the mapped discontinuities is slightly biased towards the

bottom of the window. Because of this, I would assume that the mapping was performed using traditional field methods, rather than remote sensing techniques. Either way, this should be mentioned, also to address or acknowledge the potential limitations of the methodology used, in terms, for instance, of orientation bias. Speaking of orientation bias, the fracture set 1 is suggested to be less represented – this might be due to its orientation very similar to the slope, while sets 2 and 3 are almost perpendicular. It is not clear whether this was kept into account.

Response: Thank you very much for your suggestion. When collecting fracture data in the field, the cut-off limit we considered is 1.5 m. The reason we chose this cut-off limit is that the amount of fractures with the length smaller than 1.5 m are quite large, which is beyond the artificial measurement; besides, the effect of small fractures on the slope stability is comparatively smaller than big ones. According to the cut-off limit, the number of eligible fractures is exactly 169 in the sampling window.

We are sorry for not mentioning the method we used for collecting fractures. Your assumption is right that the traditional field method, i.e., the sampling window method (Kulatilake and Wu, 1984) is used to collect fractures. The sampling window method mainly presents two limitations:1) orientation bias and 2) trace length bias. Orientation bias occurs because the probability that fractures with small intersection angles between the fractures and exposed rock surface can be collected in the field is smaller than those fractures with large angles. However, it is should be noticed that orientation bias only need to be considered when performing 3D DFN simulation (Terzaghi 1965). In the present study, a 2D analysis was performed and therefore the orientation bias can be ignored.

Trace length are biased due to two conditions: 1) only one end of a fracture is measured and (2) no end of a fracture is measured. In the present study, we corrected the trace length data using the method introduced by Kulatilake and Wu (1984). Table 1 lists the mean value and probability density function (PDF) of the corrected trace lengths for each fracture set.

In the revised manuscript, we mentioned the sampling window method and the limits of it as "Fracture characteristics, such as orientation, trace length, spacing, roughness, aperture, filling, and termination, in the exposed surface were systematically surveyed by the sampling window method" and "The sampling window method features two main limits of orientation bias and trace length bias. Orientation bias is ignored in the present study since it is only considered in performing 3D DFN simulation. The measured trace lengths bias occurs when the sampling window method is applied due to the following: (a) only one end of a fracture is measured, (b) both ends of a fracture are measured, and (c) no end of a fracture is measured".

Comment 10: Line 90: Reference here seems out of place, unless the result came from that specific work.

Response: Thank you very much for your suggestion. The reference aims to present that the grouping method used in the present study is suggested by it. It is really true that the reference should not be put in this place. In the revised manuscript, we put the reference in the right place by rewriting the sentence as "The fractures can be divided into three sets using the method proposed by Chen et al. (2005), as shown in Figure 3".

Comment 11: Line 94-95: This sentence requires rephrasing. I suggest starting from the issue of the trace length bias, and then stating the reason, rather than the contrary.

Response: Thank you very much for your suggestion. We accept your professional suggestion and rewrote this sentence as "The measured trace lengths bias occurs when the sampling window method is applied due to the following: (a) only one end of a fracture is measured, (b) both ends of a fracture are measured, and (c) no end of a fracture is measured."

Comment 12: Line 99: I recommend "and 'investigate the' potential failure mechanism".

Response: Thank you very much for your suggestion. In the revised manuscript, we

rewrote this sentence as you suggested, i.e., "The cross section normal to the exposed surface was used to perform the 2D stability analysis and investigate the potential failure mechanism of the rock slope."

Comment 13: Lines 105-107: It seems the authors suggest that the fracture intensity in a section is a function of the orientation of the set, with respect to the north. This is a bit counter-intuitive, as the orientation of the rock face (and specifically the angle with the fracture set) is surely more relevant than the azimuth (angle with the North) of the fracture – which in fact should not be that important. More detail on the method should be provided to improve clarity.

Response: We are sorry for our unclear description. The slope is oriented at a trend of 200°; we rotated the slope 20° so that the slope exactly strikes in the NS direction prior to the deduction of the function. The information above is omitted in our paper considering it has been explained in the work of Zhang et al. (2017). However, the omission of this important information obviously results in the misunderstanding. In the revised manuscript, we added this information and interpreted this function as "We rotate the slope 20° so that the slope strikes in the NS direction and assume the fracture frequency measured along the mean normal vector direction of fracture set i is $\lambda$i, and the acute angle between this direction and NS direction is $\eta$i. The fracture frequency along the line parallel to the strike of the outcrop plane is $\lambda$icos$\eta$i (Priest 1993), and the cross section plane is $\lambda$isin$\eta$i. The fracture frequency of the latter is tan$\eta$i times that of the former, and P21 (2D fracture intensity) follows this result according to the concept of the integral."

Comment 14: Line 106: P21 is "fracture intensity", not "fracture density" (which is P20 in the 2D case).

Response: We are sorry for our wrong use of the term "fracture density". It is really true that P21 is fracture intensity, which represents the length of fractures per unit area of rock mass (m/m2). P20 is fracture density, which describes the number of fractures

[Figure]

Interactive
comment

per unit area of rock mass (m-2). In the revised manuscript, we changed "2D fracture density" to "2D fracture intensity".

Comment 15: Line 117-118: The second statement seems to suggest that different input data ("fracture characteristics") were used for the four DFN realizations.

Response: We are sorry for our unclear description. "Input fracture data" is different from "fracture characteristics" in our description. The former one refers to indispensably statistical fracture data for establishing the DFN, such as the distribution types of fracture locations, P21, the mean and variance values of the trace lengths. A majority of DFNs can be generated by Monte Carlo simulation on the basis of these statistical fracture data. Therefore, input data are the same for different DFNs, which explains the first sentence "More than one DFN can be generated with the same fracture data".

The latter one represents the specific fracture characteristics that the generated DFMs present, such as the specific location, dip angle, and trace length of each facture. These fracture characteristics vary for different DFNs, which is described in the second sentence, i.e., "For example, Fig. 4 exhibits four DFN s with different fracture characteristics".

It is really true that the statements of the two sentences are misleading according to your comment; thus, we rewrote the two sentences in the revised manuscript as "More than one DFN can be generated on the basis of the aforementioned statistical fracture data. For example, Fig. 4 exhibits four DFNs with the same statistical fracture data, but fracture characteristics, such as locations, dip angles, trace lengths, are different from one another."

Comment 16: Lines 120-123: Perhaps a sketch would help the reader understanding the procedure. Also, I believe this procedure is performed in 2D. If so, I suggest to use slope "profile" instead of "surface" – this would make the procedure easier to understand for the reader (especially if no figure is provided).

Response: Thank you very much for your suggestion. It is really true that the procedure is performed in 2D. In the revised manuscript, we changed "surface" to "profile" for being easily understood.

Comment 17: Line 130: even more importantly, SRM is used to simulate the brittle propagation of fractures, and thus the brittle behavior of rock masses.

Response: Thank you very much for your comment. It is really true that SRM is widely used to simulate the brittle propagation of fractures, which we mentioned in Introduction but ignored here. In the revised manuscript, we added this application and described as "SRM approach is widely used to reproduce the mechanical properties and behaviours of fractured rock masses, simulate the fracture propagation and brittle failure of fractured rock masses, and simulate the failure and deformation of fractured rock slopes".

Comment 18: Line 134: it would be good to provide a couple of examples (even in brackets) of the micro-properties that are used as input.

Response: Thank you very much for your suggestion. It is indeed better to provide some examples of the input micro-properties first. In the revised manuscript, we added some examples of micro-properties as "The SRM model in PFC2D is defined by many parameters, such as particle contact modulus, particle normal/shear stiffness ratio, and parallel bond modulus. These parameters cannot be directly identified via laboratory and field experiments".

Comment 19: Line 160: perhaps "interpenetrate" or "compenetrate" is a better term than "pass through".

Response: Thank you very much for your suggestion. The word "interpenetrate" is indeed much better than "pass through"; thus, we replaced "pass though" with "interpenetrate" in the revised manuscript.

Comment 20: Lines 186-188: this assumption is perhaps more adequate considering

the low stress conditions that characterize the real slope.

Response: Thank you very much for your suggestion. It is really true that the investigated slope is characterized by low stress conditions; thus, we added this reason as the support of the assumption in the revised manuscript. Specifically, it is described as "This process ignored the stress concentration at the tips of the structural fractures generated by tectonic stress, which was considered reasonable in this study since the investigated slope is characterized by the low stress conditions and the stress concentration was intensely reduced after the long-term stability of the rock slope."

Comment 21: Line 190: Just a comment here. As the authors know, this approach may be "risky" in other conditions (i.e., high stress/high slopes) as it may cause a "shock" in the model, causing excessive damage in the slope, compared to a progressive excavation (or a progressive removal of the boundary), which generally is more representative of a "real world" situation.

Response: Thank you very much for your comment. It is really true this approach (one-time removal of the boundary) may cause excessive damage in the slope, especially for high slopes. However, the investigated slope was exactly formed by one excavation in the real condition; thus, the approach, i.e., one-time removal of the boundary, is practical. As for other high slopes, which may be more likely to be formed by progressive excavations, the progressive removal of the boundary is more appropriate. The specific approach to removing the boundary should be determined according to excavation methods of slopes.

Comment 22: Lines 194-197: It is unclear what are the benefits of decreasing the friction coefficient while increasing gravity. Intuitively, the gravity increase already would induce an increase in shear stresses compared to the initial state (very much like increasing the density). How is this double effect (increase in shear stress, decrease in shear strength) accounted for in the FoS calculation? And why just the friction, and not the cohesion? The paper would benefit from a more detailed explanation of the

method employed.

Response: We are sorry for our unclear description. It is really true that the increase of gravity would induce the increase in shear stress, as well as the increase in normal stress. The increases in both stresses lead to increases in driving and resisting forces, which makes the change in factor of safety unclear. Therefore, the factor of safety cannot be reflected by only increasing the gravity. Only if one of the forces (driving or resisting forces) is fixed can the change of the other be related to the factor of safety. The driving force cannot be fixed because it is directly proportional to gravity; thus, the resisting force should be fixed. The resisting force is directly proportional to the shear strength, which is equal to c+$\sigma$ tanâĄą$\varphi$ (where c is cohesion; $\sigma$ is the normal stress, and $\varphi$ is friction angle). $\sigma$ increases when the gravity increase; thus, tanâĄą$\varphi$ is considered to be reduced for making resisting force constant. In PFC, tanâĄą$\varphi$ is directly proportional to the friction coefficient of particle; thus, the decrease of the friction coefficient of particle can lead to the decrease of tanâĄą$\varphi$. In addition, the friction coefficient has little influence on cohesion; thus, making the amplitude of reduction of the friction coefficient is the same as that of the increase in gravity acceleration can ensure an approximate invariance of the resisting force. It is followed that the factor of safety is the ratio of the gravity acceleration in the limit equilibrium state (g') to that in the initial state (g), i.e., F= g' / g.

The details above are not described in the previous manuscript, which is indeed hard to tell the benefits the method. In the revised manuscript, we further interpreted the improved gravity increase method as "This method leads to the failure of a slope in PFC2D by slowly increasing gravity acceleration and reducing the friction coefficient of particles while keeping other parameters constant. Notably, the amplitude of reduction of the friction coefficient is the same as that of the increase in gravity acceleration. In this way, the resisting force can be fixed and therefore the factor of safety is directly reflected by the driving force".

Comment 23: Line 202: this seems a very stable slope. Expectedly, in view of the

amount of rock bridges along the rupture surface, which may be estimated at about 30-40%, according to figure 7.

Response: Thank you very much for your comment. It is really true that the investigated rock slope is extremely stable, which can be reflected by the high factors of safety. As you said, the amount of rock bridges along the rupture surface can also verify that the investigated slope is very stable.

Comment 24: Line 232 (and after): Perhaps it will be better to refer to the "slip mass" as "failed", or "detached" mass/volume/material.

Response: Thank you very much for your suggestion. It is really true that "failed mass" is better than "slip mass"; thus, we carefully checked all the manuscript and changed "slip mass" to "failed mass".

Comment 25: Line 255: I recommend referring to the "model geometry" or "morphology", rather than "shape".

Response: Thank you very much for your suggestion. It is really true that "morphology" is better than "shape". We carefully checked the word "shape" describing the same meaning and then replaced it with "morphology" in the revised manuscript.

Comment 26: Line 272: A variability 25-75 is indeed very high. Perhaps this variability be lower if a more realistic DFN (i.e. inclusive of smaller fractures) was to be simulated. Absolute values would be lower, for sure.

Response: Thank you very much for your comment. It is really true that the variability between factors of safety is very high. In the revised manuscript, we reduced the strength of intact materials to account for the smaller fractures, which we explained in the response to comment 1. In the recalculation of factors of safety, a lower variability is indeed observed and absolute values are also lower. We are conducting the recalculation of factors of safety of all 100 SRM model; thus, the final result is not yet available at present.

Comment 27: Line 290: again, just a comment. The limitations of this estimation is that is assumes that the base of the model is constituted by strong rock, likely with high coefficient of restitution, and the distribution of the failed mass over this distance is not considered.

Response: Thank you very much for you comment. It is really true that the base of the model is constituted by strong rock, which is represented by a rough rigid wall in PFC. The distribution of the failed mass over the distance is not analysed since this result cannot been proved a statistical significance. The accumulation results vary for 100 different SRM models, which can be verified in Fig. 13. The only thing common is that the final deposit is composed of relatively intact rock blocks and crushed particles, and the blocks pile up above the crushed particles, presenting an inverse grading phenomenon.

Comment 28: Lines 301-303: this sentence is unclear and requires rephrasing.

Response: We are very sorry for our unclear description. In the revised manuscript, we rewrote this sentence as "The factor of safety of the investigated slope is extremely high but reasonable. In the field investigation, weak interlayer and through-going discontinuities are not observed. The non-persistent fractures are very developed, which therefore play a vitally important role in the stability of the investigated slope. The safety factors of this type of slopes (i.e., slopes are characterized by non-persistent fractures) are always high".

Comment 29: Lines 303-305: I agree with the authors here: rock bridges are multiple orders of magnitude stronger than discontinuities, and this justifies the high FoS. The questions, however, is: are these estimations accurate and representative of the real situation? Figure 4 shows a rock mass significantly more fractured that the DFNs employed in this study, where the slope is formed by very large, intact blocks.

Response: Thank you very much for your comment. On the basis of our previous results (small fractures are not considered), the factors of safety are accurate and can

represent the real situation. This is because we also calculate the factors of safety by the traditional methods (i.e., the ratio of the resisting force to the driving force), which presents the same result as the simulation ones.

The DFNs in Fig. 4 are totally introduced into the simulated slopes, which is reflected by comparison between Fig. 4 and Fig.7. As for very large and intact blocks you mentioned, maybe you refer to the blocks of the boundary sections located in the bottom and right sides of the slope section. The boundary section won't affect the slope stability, which mainly contributes to overcome boundary effect. In the revised manuscript, we added the description regarding the boundary section as "The bottom and right sides of the slope section were expanded by 10 m as the boundary section, which aims to avoid boundary effect and does not affect the slope stability (Fan et al., 2004)"

Responses to comments on figures/tables

Comment 30: Figure 1c: a scale and possibly a north arrow is required

Response: Thank you very much for your suggestion. In the revised manuscript, we added the scale and the strike of the slope in Fig. 1c.

Comment 31: Figure 7: a legend bar (stress) and scales are needed for clarity

Response: Thank you very much for your suggestion. In the revised manuscript, we added the legend bar and scales in Fig. 7 as you suggested.

Comment 32: Figures 8 and 10: the use of an uniform color bar and legend would enhance the comparison of the states depicted by each sub-figure.

Response: Thank you very much for suggestion. Different color bars and legends in Figs. 8 and 10 aim to make displacements of particles clear in each pictures, which is indeed inconvenient for the comparison of different states. In the revised manuscript, we unified the color bar and legend in each sub-figure as you suggested.

Comment 33: Table 4: I recommend using the same order for micro, numerical, and

lab parameters: friction, normal, and tangential stiffness.

Response: Thank you very much for your suggestion. The same order for micro, numerical and lab parameters is more beneficial for comparing the results of parameter determination. In the revised manuscript, we changed the order of parameters to ensure they are orderly arranged.

We tried our best to improve and make changes to the manuscript. We sincerely appreciate your work and hope that our revised manuscript will be met with approval. Once again, thank you very much for your favourable comments and suggestions!

Relevant references:

Bonilla-Sierra, V., Scholtès, L., Donzé, F. V., and Elmouttie, M. K.: Rock slope stability analysis using photogrammetric data and DFN–DEM modeling, Acta Geotech., 10, 497–511, 2015.

Chen, J. P., Shi, B. F., and Wang, Q.: Study on the dominant orientations of random fractures of fractured rock mass, Chinese Journal of Rock Mechanics and Engineering, 24, 241–245, 2005.

Damjanac, B., Cundall, P.: Application of distinct element methods to simulation of hydraulic fracturing in naturally fractured reservoirs, Comput. Geotech., 71, 283-294, 2016.

Elmo, D., Stead, D., Eberhardt, E., and Vyazmensky, A.: Applications of finite/discrete element modeling to rock engineering problems, Int. J. Geomech., 13, 565–580, 2013.

Fan, S. C., Jiao, Y. Y., and Zhao, J.: On modeling of incident boundary for wave propagation in jointed rock masses using discrete element method, Comput. Geotech., 31, 57–66, 2004.

Kulatilake, P. H. S. W. and Wu, T. H.: Estimation of mean trace length of discontinuities, Rock Mech. Rock Eng., 17, 215–232, 1984.

[Figure]

Lorig, L. J., Darcel, C., Damjanac, B., Pierce, M., and Billaux, D.: Application of discrete fracture networks in mining and civil geomechanics, Mining Technology, 124, 239-254, 2015.

Priest, S. D.: Discontinuity analysis for rock engineering, London : Chapman and Hall, 1993.

Pine, R. J., Coggan, J. S., Flynn, Z. N., and Elmo, D.: The development of a new numerical modelling approach for naturally fractured rock masses, Rock Mech. Rock Eng., 39, 395–419, 2006.

Sun, S. R., Sun, H. S., Wang, Y. J., Wei, J. H., Liu, J., and Kanungo, D. P.: Effect of the combination characteristics of rock structural plane on the stability of a rock-mass slope, B Eng Geol Environ., 73, 987–995, 2014.

Terzaghi, K.: Stability of steep slopes on hard unweathered rock, Géotechnique, 12, 251–270, 1962.

Zhang, W., Zhao, Q. H., Chen, J. P., Huang, R. Q., and Yuan, X. Q.: Determining the critical slip surface of a fractured rock slope considering preexisting fractures and statistical methodology, Landslides, 14, 1253–1263, 2017.

---

## Author Response (AR1)

**Dear Editor and Referees:**

Thank you very much for your valuable comments and suggestions concerning our manuscript entitled "Stability evaluation and potential failure process of rock slopes characterized by non-persistent fractures" (ID: nhess-2020-58). These comments are all valuable and helpful for revising and improving our paper, as well as the important guiding significance to our researches. Careful revisions have been made to the manuscript according to the comments and suggestions, which we hope will merit your approval. The revised portions are marked in red in the manuscript. A point-by-point response to your comments and relative changes made in the manuscript are listed as follows.

**Responses to Editor**

Comment 1: on the basis of the reports of two peer-reviewers, complemented by my own revision of your manuscript, I consider that the content of your research may be of interest for the journal reader and may represent a suitable publication in NHESS should you be ready to incorporate some major revisions.

**Response**: Thank you very much for your positive evaluation. We have revised our manuscript carefully according to all comments and the revised portions are marked in red in the revised manuscript. We sincerely hope our revised paper will merit your approval.

*Comment 2:* Please carefully look at all the comments and remarks made by the two referees and reply to them, one by one. Where required, please modify the original manuscript and provide a new amended copy along with the document with replies to the reviews.

**Response**: Thank you very much for your comment. We have carefully looked at and replied to all comments and suggestions one by one, which can be found in Responses to Referee #1 and Responses to Referee #2. In addition, the previous manuscript have been modified according to these comments and suggestions, which can be found in the relative responses as well as the revised manuscript.

**Comment 3:** In particular, I am concerned about the main issue that is raised by Reviewer #2 on

the model adopted to represent the rock mass and the fracture network. I agree with the reviewer that such DFN representation, despite being suitable in terms of conceptual model representation, does not seem suitable for the representation of the actual rock mass, where the role of the small fracture sub-network cannot be overlooked without compromising the entire numerical description of the whole.

**Response:** Thank you very much for your comment. It is really true that the role of the small fractures cannot been overlooked in the stability analysis. DFNs in the previous manuscript only consider the fractures with trace lengths larger than 1.5 m, which indeed overestimate the stability of the rock slope. In the revised manuscript, we accepted the suggestion of Referee #2, i.e., decreasing the strength of intact rock materials to account for the small fractures that are not implemented as discrete discontinuities in the DFN. The new numerical results demonstrated that the new safety of factors (12-38) were lower than the previous ones (25-73.5), while the critical slip surface and potential failure process remained the same as the previous ones. Relative changes in the revised manuscript have listed in the responses to Referee #2 and therefore are not described in detail in this place.

Comment 4: Also, please consider the remarks of reviewer #1, with specific reference to the one related to the choice of the slope for the numerical analysis, which is quite stable and not prone to failure. That leaves no room to variability in slope behaviour and seems to underestimate the scope of the work. Again, this issue could be related to the previous one as raised by reviewer #2.

**Response**: Thank you very much for your comment. The comprehensive method we proposed aims to analyse the stability of rock slopes characterized by non-persistent fractures. Therefore, we selected a slope whose stability is completely controlled by non-persistent fractures. However, the safety factors of this type of slopes are always high (more than 10) due to the high strength of rock bridges (Huang et al., 2015). Although the slope we analysed is quite stable and failure hardly occur in the future, we believe the proposed method really make sense for engineering projects related to fractured rock slopes.

*Comment 5:* Please also consider all remaining remarks and minor requested made by the two referees, as in the attached material. I am confident that, after considering all the previous points,

**your manuscript will be in a form which will be of great interest for the journal readers.**

**Response**: Thank you very much for your suggestion. We have looked at all comments and remarks made by the two referees one by one and modified our manuscript based on these comments and remarks. The revised portions are marked in red in the revised manuscript. Meanwhile, relative revisions are also listed in the corresponding responses to Referee #1 and Referee #2.

We tried our best to improve and make changes to the manuscript. We sincerely appreciate your work and hope that our revised manuscript will be met with approval. Once again, thank you very much for your favorable comments and suggestions!

**Relevant references:**

Huang, D., Cen, D. F., Ma, G. W., and Huang, R. Q.: Step-path failure of rock slopes with intermittent joints, Landslides, 12, 911–926, 2015.

**Responses to Referee #1**

*Comment 1:* L 69-70: Authors state: "These slopes may become hidden dangers (e.g., geological disasters) and pose potential threats to people and nearby equipment". This sentence is in strong contrast with the results from the stability analyses, since the safety factors range from 25 to 75. I'm wondering if the investigated slope is really too stable or the safety factors could have been somehow overestimated. Nevertheless, no event seems to have occurred in recent times from the investigated slope; this is a pity, since real events could have been extremely useful for validating the performed analyses. Why did Authors choose to test the proposed procedure to this slope? *Response:* Thank you very much for your comment. It is really true that the investigated slope is stable according to its safety factors; thus, a large-scale rock slide can hardly occur. Only some small-scale rock falls may happen. Therefore, we corrected this sentences as "These slopes may become rock falls and pose potential threats to people and nearby equipment. Whether rock slide will happen requires calculating and evaluating". (Page 3: Lines 71-73 in the revised manuscript)

The investigated rock slope is highly stable according to the following reasons: 1) weak interlayers and through-going discontinuities are not developed in the field; 2) the amount of rock

bridges along the slip surface is about 30%-40%, which will stabilize the slope because the rock bridges are multiple orders of magnitude stronger than fractures.

It is really a pity since no real event can be used for validating the performed analyses. We have investigated many destroyed slopes. The results shown that the stabilities of the slopes are always controlled by weak interlayers or through-going discontinuities. This type of slopes is disadvantageous to our analysis of the influence of non-persistent fractures on rock slope stability. Therefore, we selected a slope whose stability is completely controlled by non-persistent fractures. However, the safety factors of this type of slopes are always high (more than 10) (Huang et al., 2015). In the revised manuscript, we decreased the strength of the intact rock materials according to equivalent strength of equivalent rock mass. The new modeling results showed the factors of safety remains high (from 12 to 38) but lower than the previous ones (from 25 to 73.5). We will select the rock slope with more fractures to test the proposed method for future researches. Nevertheless, we believe the proposed method really make sense for engineering projects related to fractured rock slopes.

Comment 2: L 92-93 and figure 3: the division of the detected discontinuities into sets is rather questionable, since data dispersion is indeed too high. Yet, this should not affect model reconstruction too much, as dispersion is also considered in the generation of artificial discontinuities within the DFN model.

**Response:** Thank you very much for your comment. It is really true that the data dispersion is high. An important reason of this phenomenon is that small-scale structural fractures with highly dispersed orientations predominate in the exposed rock surface. Subsequently, some fracture poles are also dispersed for each set. However, the distances between these fracture poles and the centre of the set is indeed closer compared with that between them and the other two sets, which was validated using the method proposed by Chen et.al (2005). Therefore, the grouping results are reasonable.

It is really true as you said that the data dispersion is high, but it is not caused by grouping. Even though fractures with similar orientations are divided into one set, the orientations of fractures in the same set still present dispersion to some extent. We considered the dispersion of all fracture orientations in the field. The final DFN model is formed by combining three fracture sets. Therefore, the data dispersion will not affect the establishment of the DFN model.

**Comment 3:* L184: I suggest not to take for granted what a "fish function" is; please explain for those unfamiliar with numerical modelling.**

**Response**: Thank you very much for your suggestion. It is really true that fish function may be hard to be understood for those unfamiliar with numerical modelling. Therefore, "fish function" was interpreted in the revised manuscript as "Then, an embedded scripting language in PFC, i.e., FISH, is used to write user-defined functions for extending the functionality or adding user-defined features in PFC. In the present study, we used the FISH functions to add the DFN into the model of the slope section by reading the location data of fractures. Subsequently, the SRM model composed of the BPM and DFN is established". (Page 8: Lines 231-234)

Comment 4: L255-259: The modelling results are nice and look quite reasonable, but without validation they seem to be an end into themselves. Did any real event occur from the investigated slope in recent times? Are there any detached blocks to compare their volume and shapes with the simulated ones? What about their runout? In my opinion the reliability of the proposed procedure needs to be demonstrated.

**Response**: Thank you very much for your positive feedback of the modelling results. No rock slide has occurred in recent times from the investigated slope, which is clearly confirmed by the high factors of safety computed in the simulations; thus, it is hard to compare the volume and shapes of the real and simulated ones. It is really a pity since no real event can be used for validating the proposed procedure. We will select the rock slope with more fractures to test the proposed method for future researches.

*Comment 5:* L260: Please change section 6 title, since "stability analysis" has already been used for section 4. Maybe, "Statistical analysis"?

**Response**: We are very sorry for our carelessness. It is really true that the title of section 6 should be "Statistical analysis". In the revised manuscript, the title of section 6 has been changed to "Statistical analysis". (Page 10: Line 311)

**Comment 6: L281: please change are 43.5 to is 43.5**

**Response**: We are very sorry for our improper word. In the revised manuscript, we changed "are" to "is" (Page 11: Line 332). In addition, we carefully checked the revised manuscript to ensure that all vocabulary and grammar errors were corrected.

**Comment 7:** L327 and 334: To trigger instability, Authors used the gravity increase method proposed by Meng et al. (2015). It is not clear to me if Authors made some amendments to this method, or if they used it as is. If so, in my opinion, the method cannot be defined as "innovative". *Response:* Thank you very much for your suggestion. In the present research, we directly applied the improved gravity increase method proposed by Meng et al. (2015) to trigger instability. We are very sorry for our wrong use of the word "innovative". Although this method is relatively new compared with traditional gravity increase method, this study cannot refer this approach as an innovative one. Therefore, we deleted the word "innovative" and rewrote this sentence as "The factor of safety is determined on the basis of the improved gravity increase method". (Page 12: Lines 377-378)

*Comment 8:* Figure 1: Fig 1b: image is not clear; I suggest deleting the text superimposed over the image. Fig 1c: if the image contains the window reported in Figure 2, please add the box limits. *Response:* Thank you very much for your suggestion. In the revised manuscript, we deleted the text superimposed over the image in Fig. 1b as you suggested. In addition, we enhanced the

contrast to improve the image clarity. (Page 17: Figure 1b)

It is really true that Fig. 1c contains the window reported in Fig. 2; thus, as you suggested, we added the box limits corresponding to the sampling window in Fig. 1c. (Page 17: Figure 1c)

*Comment 9:* Figure 7 In my opinion, the picture in picture representation is misleading. If possible, add all the drawings in a single image, otherwise, split in two different figures.

**Response**: Thank you very much for your suggestion. It is really true that one image composed of two pictures is misleading. In the revised manuscript, we added all drawings in a single image. In addition, we added detailed description in the caption of Fig. 7, which would help further understand the meaning of each image. (Page 20: Figure 7)

Comment 10: Figure 8: to improve image resolution, I suggest cropping the images to the

**fractured sectors.**

**Response**: Thank you very much for your suggestion. It is really true that the resolution of Fig.8 is low. In the revised manuscript, the boundary sector without fractures were cropped and only remained the fractured sectors to improve image resolution. (Page 21: Figure 8)

Comment 11: Figure 9: sometimes the timestep count is represented in decimal notation and sometimes in scientific notation, please uniform.

**Response**: Thank you very much for your suggestion. It is really true that the expression of the timestep count should be uniformed. In the revised manuscript, we changed the timestep count represented in decimal notation to scientific notation. Specifically, 2000, 5000, 10000, 20000, 40000, 60000, and 80000 were changed into  $2 \times 10^3$ ,  $5 \times 10^3$ ,  $10^4$ ,  $2 \times 10^4$ ,  $4 \times 10^4$ ,  $6 \times 10^4$ , and  $8 \times 10^4$ , respectively. (Page 20: Figure 7; Page 21: Figure 8)

We thank you for your valuable comments and suggestions. These comments are all valuable and helpful in revising and improving our paper, as well as in guiding the significance of our research.

**Relevant references:**

Chen, J. P., Shi, B. F., and Wang, Q.: Study on the dominant orientations of random fractures of fractured rock mass, Chinese Journal of Rock Mechanics and Engineering, 24, 241–245, 2005. Huang, D., Cen, D. F., Ma, G. W., and Huang, R. Q.: Step-path failure of rock slopes with intermittent joints, Landslides, 12, 911–926, 2015.

Meng, Y. D., Su, Q. M., Lu, W. P., and Xu, Z.: Research on improved grain flow gravity increase method, Chinese Journal of Water Resource and Power, 33, 149–151, 2015.

**Responses to Referee #2**

Comment 1: The first general comment is that the investigated slope is quite low (20 m in the simulated models). It is true that in certain conditions brittle damage can develop even in low stress conditions, especially when tensile strength is exceeded or in case of significant stress

concentrations. However, in this case it seems that the amount of rock bridges along the critical path(s), would most likely stabilize the slope, except perhaps for small blocks at surface. It is very unlikely that the real slopes will ever fail with the simulated mechanism, unless smaller fractures are included in both the data collection and DFN. This is clearly confirmed by the high factors of safety computed in the simulations.

This brings me to the second, important comment: the model input data, which seems the "weak point" of this manuscript. It is not clear what technique was used for the initial data collection, but the DFN used in the models seems to rely on and include only the larger discontinuities. This results obvious by visually comparing figure 1c and figure 2 (which seems to depict a much more fractured rock mass). Considering the input DFN, the high factors of safety computed makes sense. However, the question is: "is this DFN a realistic representation of the real rock mass?". The issue with the input data may have significantly impacted the numerical results, in terms of factor of safety, and location of the critical path. A "work around" would be to decrease the simulated strength of the intact material, to account for the smaller fractures that are not implemented as discrete discontinuities in the DFN.

In conclusion, in its present form, the manuscript seems quite conceptual, in that there is a seemingly weak connection between the model and the actual rock mass. While the simulations may indeed reproduce a realistic fracture propagation mechanism, the actual process is very unlikely to occur in the simulated slope, or any slope with similar height, lithology, and structural configuration. In view of this, this paper should be significantly improved (major revisions) with regard to the model input data and their description. Either by a) including the fractures that seem to be missing in the DFN, or b) demonstrating in a clearer manner the similarity between the DFN and the rock mass. This could include better pictures, with close-ups and, importantly, scales. Showing the mapped traces onto the photo of the slope could also greatly improve the clarity.

**Response**: Thank you very much for your comment. It is really true that the investigated slope can hardly fail considering the low height, a large amount of rock bridges along the slip surface. This is also confirmed by the high factors of safety. When it is subjected to significant environmental changes, such as earthquakes, rainfall, unloading, or overloading, the failure may occur following

the potential failure mechanism.

The sampling window method (Kulatilake and Wu 1984) is used to collect fracture data in the present study. It is really true that the generated DFNs rely on and include the larger fractures since only fractures with the length larger than 1.5 m were measured in the field. The amount of fracture with the length smaller than 1.5 m are extremely large, which goes beyond the artificial measurement. Considering that small fractures have a little effect on slope stability, we take the cut-off limit of 1.5 m.

On the basis of the collected fracture data in the exposed rock surface, the DFN is a most possible representation of the real rock mass. However, it is really true that the DFN is more accurate and the numerical results are more reasonable if smaller factures are taken into consideration. In the revised manuscript, we decreased the strength of the intact material to remedy the lack of smaller fractures in DFN. Specifically, particle parameters which influence the strength of intact material, including the friction coefficient of particles, tensile strength of parallel bond, and cohesion of parallel bond are synchronously reduced by half according to the equivalent shear strength of equivalent rock mass.

The new modelling results demonstrated that the factors of safety are lower when the strength of intact rock material than the previous ones, indicating the lack of smaller fractures indeed makes the factors of safety relatively high. Even so, the factors of safety remain high, which are between 12 and 38. In addition, the location of the critical slip surface remains the same as the previous one since the slip surface is always composed of pre-existing fractures and new-propagated ones. Similarly, the potential failure process is roughly identical to that of previous simulation since both are conducted in the natural condition (i.e., the gravity acceleration is 10.0).

In the revised manuscript, corresponding content of parameter reduction and the new modelling results are rewrote. Details can be found in Lines 90-92 and 176-204 in the revised manuscript, which are also listed in the end of this response. In addition, the scales were added in all required figures and the location of the mapping window has been added in Fig. 1c. We also carefully checked the whole manuscript and considered all of your line-by-line comments.

Page 3: Lines 90-92: "Fractures in the exposed surface with trace lengths smaller than 1.5 m are widely distributed and difficult to record. Therefore, these fracture are not considered when performing fracture data collection for 2D DFN simulation. Nevertheless, the effect of these fractures on the rock mass strength is considered, which is explained in Sect. 3.1".

Pages 6-7: Lines 176-204: "It should be noted that parameters in Table 2 are representations of intact rock materials in PFC2D. However, fractures with trace lengths smaller than 1.5 m were disregarded in the collection of fracture data, but these clearly represent a weakness. Considering the effect of these fractures on strength of the intact rock material make a significant difference to the following stability analysis. In the present study, the combination of intact rock mass and these small fractures is considered as the equivalent rock mass. Obtaining the equivalent shear strength of the equivalent rock mass failure is generally promoted by the reduction of the shear strength of rock mass (i.e., the shear strength reduction method). An equivalent shear strength calculation was developed based on the Mohr-Coulomb criterion (Lajtai, 1969b; Shang et al., 2018), as expressed by the following equation:

 $\tau = c_e + \sigma \tan \varphi_e = [K_L \cdot c_f + (1 - K_L) \cdot c_R] + \sigma [K_L \cdot \tan \varphi_f + (1 - K_L) \cdot \tan \varphi_R]$  (1) where  $\tau$  and  $\sigma$  represent equivalent shear strength of the equivalent rock mass and normal stress;  $c_e$  and  $\varphi_e$  are the equivalent cohesion and friction angle of equivalent rock mass;  $c_f$  and  $\varphi_f$ are the cohesion and friction angle of fractures;  $c_R$  and  $\varphi_R$  are the cohesion and friction angle of intact rock mass;  $K_L$  is the linear persistence.

The values of  $c_R$ ,  $\varphi_R$  and  $\varphi_f$  are 12.25, 25, and 18, which are listed in Tables 2 and 3. Filed investigation demonstrated that no fillings existed in fractures, implying the cohesion of fractures is equal to zero (i.e.,  $c_f = 0$ ). The linear persistence is defined as the ratio of fracture trace lengths and the total length of coplanar given line (Shang et al., 2018; Zhang et al., 2020b). In the present study, several lines with different directions are set in the exposed surface and then the linear persistence is measured. The average linear persistence is considered as the final linear persistence of the equivalent rock mass, whose value is around 50%. Subsequently, substituting aforementioned parameters into Eq. (1), we can deduce that the shear strength of the equivalent rock mass is slightly larger than half of intact rock mass. Nevertheless, Eq. (1) tends to overestimate the shear strength of equivalent rock mass; thus, the shear strength of the equivalent rock mass is assumed as half of intact rock mass. This assumption is beneficial to the equivalent reduction of relative parameters of intact rock mass in PFC2D; simultaneously, a smaller strength of equivalent rock mass contributes to a relatively small factor of safety, which is more conservative and favourable for engineering projects.

The shear strength of rock materials is controlled by three parameters (including tensile strength of parallel bond, cohesion of parallel bond and friction coefficient of particles) in PFC2D (Bonilla-Sierra et al., 2015). Therefore, these three parameters are synchronously reduced by half while other parameters are kept constant. Specifically, the values of the tensile strength and cohesion of parallel bond and friction coefficient of particles are 12.5MPa, 12.5MPa and 0.35, respectively. By this way, the effect of small fractures are considered in the generation of intact rock mass. Equivalent parameters of the equivalent rock material are obtained and adopted in the generation of SRM model."

Notably, the contents of critical slip surface and potential failure process are not modified because they are the same as the previous ones. The only difference is that the values of factors of safety and thus only this portion is changed.

*Comment 2:* Line 40: authors should refer to "the non-persistent fractures in these works", rather than "these non-persistent fractures".

**Response:** Thank you very much for your correction. In the revised manuscript, we have changed "these non-persistent fractures" to "the non-persistent fractures in these works". (Page 2: Line 40)

*Comment 3:* Line 51: SRM has been used also for underground applications, including mining and hydraulic fracturing.

**Response**: Thank you very much for your comment. We carefully searched the literatures regarding underground applications of SRM approach. It is really true that SRM approach has been widely used in mining and hydraulic fracturing. Therefore, we added corresponding description as "SRM models have been primarily used to simulate failure and deformation of fractured rock slopes (Bonilla-Sierra et al., 2015; Elmo et al., 2013), simulate hydraulic fracturing in naturally fractured reservoirs (Damjanac and Cundall, 2016), and estimate rock mass strength,

fragmentation and micro seismicity in caving mines (Lorig et al., 2017)". (Page 2: Lines 50-53)

*Comment 4:* Lines 51-53: perhaps these two sentences can be merged. However, the first sentence requires rephrasing, as it seems some words are missing.

**Response**: Thank you very much for your suggestion. The latter sentence is the explanation of the former one; thus, these two sentences can indeed be merged. In the revised manuscript, we merged the two sentences as "DFN simulation included in SRM modeling program presents a significant variability, which means numerously possible realizations of 2D fracture systems exist given specified input parameters (Pine et al., 2006; Zhang et al., 2020a)". (Page 2: Lines 53-55)

*Comment 5:* Line 58: authors should state the country the investigated site is located in.

**Response**: Thank you very much for your suggestion. In the revised manuscript, we stated the country the investigated site is located in and described as "This study proposes a comprehensive approach that combines several well-established methods to conduct a stability evaluation and failure process analysis of a fractured rock slope in Tianjin City, China." (Page 2: Line 61)

*Comment 6:* Line 71-72: this sentence can be improved. Perhaps the slopes are higher to the south, rather than the quarry area itself. Also, from this sentence it is not clear whether the "mountain "is a ridge oriented north-south, or the if the bedding are dipping to north (or south).

**Response**: We are very sorry for our unclear description. Actually, all sentences from Line 71 to 73 contribute to the description of the study area (i.e., the quarry area). Therefore, it is true that the quarry area is higher in the north. The "mountain" in this place refers to the monoclinal mountains striking south-north. In the revised manuscript, we rewrote these sentences as "The Laohuding Quarry area is characterized by the low-mountain terrain, which is higher in the north than in the south. The highest and lowest altitudes of the quarry area are 160 m and 60 m, with a relative elevation of 100 m. A majority of monoclinal mountains striking south–north exist in this area. The average slopes of the mountains in the east and west of the quarry area are  $25^{\circ}$  and  $30^{\circ}$ , respectively (Fig. 1b)". (Page 3: Lines 74-77)

*Comment 7:* Line 79: is the formation thick, or the limestone layers? Either way, how thick? Karst phenomena are not obvious, meaning there is not any, or that they are not or scarcely visible? *Response:* We are very sorry for unclear description. Actually, we initially aims to say "The

limestone is moderately weathered" rather than "The limestone is moderately thick". In the revised manuscript, we corrected it. (Page 3: Line 83) "the karst phenomena are not obvious" means that the karst phenomena are scarcely visible in the study area due to low precipitation and the lack of groundwater. In the revised manuscript, we rewrote this sentence as "the karst phenomena are scarcely visible due to low precipitation and groundwater shortage." (Page 3: Lines 83-84)

Comment 8: Lines 80-83: the last two sentences could be merged. The term "intermittent" for discontinuities is somewhat inaccurate (or simply very rarely used, to my knowledge) – perhaps simply "non-persistent" is more appropriate. Additionally, one would expect that bedding would be very persistent. Will this play a role? Although it is true that observing the bedding trace does not necessarily imply a fully persistent plane with no tensile strength.

**Response:** Thank you very much for your suggestion. It is really true that the term "intermittent" for discontinuities is rarely used. In the revised manuscript, we used "non-persistent" to substitute it as you suggested. (Page 3: Line 85) Field observation demonstrated that no bedding planes, faults, folds, and shear zones are developed in the rock exposure. Therefore, the bedding plays no role in stability analysis. It is non-persistent fractures that play the most significant role in the slope stability and potential failure process. Aforementioned descriptions are rewrote in the revised manuscript as "Faults, folds, bedding planes, shear zones and weak interlayers are not observed, and thus, this area is tectonically stable. Non-persistent discontinuities are randomly and widely developed in outcrops (Fig. 1c). Therefore, it is the non-persistent discontinuities (fractures) that control the slope stability and potential failure process". (Page 3: Lines 84-87)

Comment 9: Line 86-92: A 62 by 6 m is a large area to perform systematic discontinuity mapping (i.e. using traditional field techniques or short range remote sensing methods), and 169 discontinuities seems a low figure – what is the cut-off limit you considered (i.e. the smallest fracture that was considered). Looking at figure 2, it seems that the location of the mapped discontinuities is slightly biased towards the bottom of the window. Because of this, I would assume that the mapping was performed using traditional field methods, rather than remote sensing techniques. Either way, this should be mentioned, also to address or acknowledge the

potential limitations of the methodology used, in terms, for instance, of orientation bias. Speaking of orientation bias, the fracture set 1 is suggested to be less represented – this might be due to its orientation very similar to the slope, while sets 2 and 3 are almost perpendicular. It is not clear whether this was kept into account.

**Response**: Thank you very much for your suggestion. When collecting fracture data in the field, the cut-off limit we considered is 1.5 m. The reason we chose this cut-off limit is that the amount of fractures with the length smaller than 1.5 m are quite large, which is beyond the artificial measurement; besides, the effect of small fractures on the slope stability is comparatively smaller than big ones. According to the cut-off limit, the number of eligible fractures is exactly 169 in the sampling window.

We are sorry for not mentioning the method we used for collecting fractures. Your assumption is right that the traditional field method, i.e., the sampling window method (Kulatilake and Wu, 1984) is used to collect fractures. The sampling window method mainly presents two limitations:1) orientation bias and 2) trace length bias. Orientation bias occurs because the probability that fractures with small intersection angles between the fractures and exposed rock surface can be collected in the field is smaller than those fractures with large angles. In our work, the fractures were divided into three sets, namely, fractures of sets 1, 2, and 3 with average orientations of 39.5 987.3 °(dip direction/dip angle), 307.4 944.7 °, and 110.2 931.7 °, respectively. The strikes of sets 2 and 3 are similar and almost 90 degrees apart from that of set 1. The modelled section, i.e., the cross section, is normal to the exposed rock surface, indicating the strike of it is 110 degrees and is similar to that of set 1 (The strike and dip direction are 90 degrees apart). Therefore, a substantial portion of the fractures of set 1 are not be used in the simulation of 2D DFN models and in the following stability analysis.

Trace length are biased due to two conditions: 1) only one end of a fracture is measured and (2) no end of a fracture is measured. In the present study, we corrected the trace length data using the method introduced by Kulatilake and Wu (1984). Table 1 lists the mean value and probability density function (PDF) of the corrected trace lengths for each fracture set.

In the revised manuscript, we mentioned the sampling window method and the limits of it as

"The characteristics (such as orientation, trace length, spacing, roughness, aperture, filling, and termination) of fractures with trace lengths larger than 1.5 m were systematically surveyed using the sampling window method (Kulatilake and Wu, 1984)" (Page 3: Lines 92-94), "The sampling window method features two main limits of orientation bias and trace length bias. The measured trace lengths bias occurs due to the following: (a) only one end of a fracture is measured, (b) both ends of a fracture are measured, and (c) no end of a fracture is measured" (Page 4: Lines 103-105), and "Orientation bias occurs because the fractures with small intersection angles between the fractures and exposed rock surface are more possibly collected in the field than those fractures with large angles. In the present study, the cross section normal to the exposed surface was used to perform the 2D stability analysis. The dip directions of fractures in set 1 are similar to that of the cross section, implying fractures acted as the surface of separation, which will not influence the results of stability analysis. Therefore, a substantial portion of the fractures of set 1 (fractures intersected by the cross section with an angle smaller than 20 °) were artificially deleted prior to the stability analysis". (Page 4: Lines 107-113)

*Comment 10:* Line 90: Reference here seems out of place, unless the result came from that specific work.

**Response:** Thank you very much for your suggestion. The reference aims to present that the grouping method used in the present study is suggested by it. It is really true that the reference should not be put in this place. In the revised manuscript, we put the reference in the right place by rewriting the sentence as "The fractures can be divided into three sets using the method proposed by Chen et al. (2005), as shown in Figure 3". (Page 4: Lines 98-99)

*Comment 11:* Line 94-95: This sentence requires rephrasing. I suggest starting from the issue of the trace length bias, and then stating the reason, rather than the contrary.

**Response**: Thank you very much for your suggestion. We accept your professional suggestion and rewrote this sentence as "The sampling window method features two main limits of orientation bias and trace length bias. The measured trace lengths bias occurs due to the following: (a) only one end of a fracture is measured, (b) both ends of a fracture are measured, and (c) no end of a fracture is measured". (Page 4: Line 103-105)

**Comment 12: Line 99: I recommend "and 'investigate the' potential failure mechanism".**

**Response**: Thank you very much for your suggestion. In the revised manuscript, we rewrote this sentence as you suggested, i.e., "The cross section normal to the exposed surface was used to perform the 2D stability analysis and investigate the potential failure mechanism of the rock slope". (Page 4: Lines 109-110)

Comment 13: Lines 105-107: It seems the authors suggest that the fracture intensity in a section is a function of the orientation of the set, with respect to the north. This is a bit counter-intuitive, as the orientation of the rock face (and specifically the angle with the fracture set) is surely more relevant than the azimuth (angle with the North) of the fracture – which in fact should not be that important. More detail on the method should be provided to improve clarity.

**Response:** We are sorry for our unclear description. The slope is oriented at a trend of 200 °, we rotated the slope 20 ° so that the slope exactly strikes in the NS direction prior to the deduction of the function. The information above is omitted in our paper considering it has been explained in the work of Zhang et al. (2017). However, the omission of this important information obviously results in the misunderstanding. In the revised manuscript, we added this information and interpreted this function as "We rotate the slope 20° so that the slope strikes in the NS direction and assume the fracture frequency measured along the mean normal vector direction of fracture set *i* is  $\lambda_i$ , and the acute angle between this direction and NS direction is  $\eta_i$ . The fracture frequency along the line parallel to the strike of the outcrop plane is  $\lambda_i \cos \eta_i$  (Priest 1993), and the cross section plane is  $\lambda_i \sin \eta_i$ . The fracture frequency of the latter is  $\tan \eta_i$  times that of the former, and  $P_{21}$  (2D fracture intensity) follows this result according to the concept of the integral". (Page 4: Lines 118-122)

**Comment 14:* Line 106: P21 is "fracture intensity", not "fracture density" (which is P20 in the 2D case).**

**Response**: We are sorry for our wrong use of the term "fracture density". It is really true that  $P_{21}$  is fracture intensity, which represents the length of fractures per unit area of rock mass (m/m2).  $P_{20}$  is fracture density, which describes the number of fractures per unit area of rock mass (m-2). In the revised manuscript, we changed "2D fracture density" to "2D fracture intensity". (Page 4: Lines

**121-122)**

Comment 15: Line 117-118: The second statement seems to suggest that different input data ("fracture characteristics") were used for the four DFN realizations.

**Response**: We are sorry for our unclear description. "Input fracture data" is different from "fracture characteristics" in our description. The former one refers to indispensably statistical fracture data for establishing the DFN, such as the distribution types of fracture locations,  $P_{21}$ , the mean and variance values of the trace lengths. A majority of DFNs can be generated by Monte Carlo simulation on the basis of these statistical fracture data. Therefore, input data are the same for different DFNs, which explains the first sentence "More than one DFN can be generated with the same fracture data".

The latter one represents the specific fracture characteristics that the generated DFMs present, such as the specific location, dip angle, and trace length of each facture. These fracture characteristics vary for different DFNs, which is described in the second sentence, i.e., "For example, Fig. 4 exhibits four DFN s with different fracture characteristics".

It is really true that the statements of the two sentences are misleading according to your comment; thus, we rewrote the two sentences in the revised manuscript as "More than one DFN can be generated on the basis of the aforementioned statistical fracture data. For example, Fig. 4 exhibits four DFNs with the same statistical fracture data, but fracture characteristics, such as locations, dip angles, trace lengths, are different from one another". (Page 5: Lines 132-134)

Comment 16: Lines 120-123: Perhaps a sketch would help the reader understanding the procedure. Also, I believe this procedure is performed in 2D. If so, I suggest to use slope "profile" instead of "surface" – this would make the procedure easier to understand for the reader (especially if no figure is provided).

**Response**: Thank you very much for your suggestion. It is really true that the procedure is performed in 2D. We are sorry for the wrong use of "surface". In the revised manuscript, we changed "the exposed surface" to "lines extending along the dip direction of the cross section". (Page 5: Lines 138)

Comment 17: Line 130: even more importantly, SRM is used to simulate the brittle propagation of

**fractures, and thus the brittle behavior of rock masses.**

**Response**: Thank you very much for your comment. It is really true that SRM is widely used to simulate the brittle propagation of fractures, which we mentioned in Introduction but ignored here. In the revised manuscript, we added this application and described as "SRM approach is widely used to reproduce the mechanical properties and behaviours of fractured rock masses, simulate the fracture propagation and brittle failure of fractured rock masses, and perform stability analysis of fractured rock slopes". (Page 5: Lines 146-148)

*Comment 18:* Line 134: it would be good to provide a couple of examples (even in brackets) of the micro-properties that are used as input.

**Response**: Thank you very much for your suggestion. It is indeed better to provide some examples of the input micro-properties first. In the revised manuscript, we added some examples of micro-properties as "The SRM model in PFC2D is defined by many parameters, such as particle contact modulus, particle normal/shear stiffness ratio, and parallel bond modulus. These parameters cannot be directly identified via laboratory and field experiments". (Page 5: Lines 150-151)

*Comment 19:* Line 160: perhaps "interpenetrate" or "compenetrate" is a better term than "pass through".

**Response**: Thank you very much for your suggestion. The word "interpenetrate" is indeed much better than "pass through"; thus, we replaced "pass though" with "interpenetrate" in the revised manuscript. (Page 7: Line 206)

*Comment 20:* Lines 186-188: this assumption is perhaps more adequate considering the low stress conditions that characterize the real slope.

**Response**: Thank you very much for your suggestion. It is really true that the investigated slope is characterized by low stress conditions; thus, we added this reason as the support of the assumption in the revised manuscript. Specifically, it is described as "This process ignored the stress concentration at the tips of the structural fractures generated by tectonic stress, which was considered reasonable in this study since the investigated slope is characterized by the low stress conditions and the stress concentration was intensely reduced after the long-term stability of the

rock slope". (Page 8: Lines 235-238)

Comment 21: Line 190: Just a comment here. As the authors know, this approach may be "risky" in other conditions (i.e., high stress/high slopes) as it may cause a "shock" in the model, causing excessive damage in the slope, compared to a progressive excavation (or a progressive removal of the boundary), which generally is more representative of a "real world" situation.

**Response**: Thank you very much for your comment. It is really true this approach (one- time removal of the boundary) may cause excessive damage in the slope, especially for high slopes. However, the investigated slope was exactly formed by one excavation in the real condition; thus, the approach, i.e., one-time removal of the boundary, is practical. As for other high slopes, which may be more likely to be formed by progressive excavations, the progressive removal of the boundary is more appropriate. The specific approach to removing the boundary should be determined according to excavation methods of slopes.

**Comment 22:** Lines 194-197: It is unclear what are the benefits of decreasing the friction coefficient while increasing gravity. Intuitively, the gravity increase already would induce an increase in shear stresses compared to the initial state (very much like increasing the density). How is this double effect (increase in shear stress, decrease in shear strength) accounted for in the FoS calculation? And why just the friction, and not the cohesion? The paper would benefit from a more detailed explanation of the method employed.

**Response:** We are sorry for our unclear description. It is really true that the increase of gravity would induce the increase in shear stress, as well as the increase in normal stress. The increases in both stresses lead to increases in driving and resisting forces, which makes the change in factor of safety unclear. Therefore, the factor of safety cannot be reflected by only increasing the gravity. Only if one of the forces (driving or resisting forces) is fixed can the change of the other be related to the factor of safety. The driving force cannot be fixed because it is directly proportional to gravity; thus, the resisting force should be fixed. The resisting force is directly proportional to the shear strength, which is equal to  $c + \sigma \tan \varphi$  (where *c* is cohesion;  $\sigma$  is the normal stress, and  $\varphi$  is friction angle).  $\sigma$  increases when the gravity increase; thus,  $\tan \varphi$  is considered to be reduced for making resisting force constant. In PFC,  $\tan \varphi$  is directly proportional to the friction coefficient of

particle; thus, the decrease of the friction coefficient of particle can lead to the decrease of  $\tan \varphi$ . In addition, the friction coefficient has little influence on cohesion; thus, making the amplitude of reduction of the friction coefficient is the same as that of the increase in gravity acceleration can ensure an approximate invariance of the resisting force. It is followed that the factor of safety is the ratio of the gravity acceleration in the limit equilibrium state (g') to that in the initial state (g), i.e., F = g' / g.

The details above are not described in the previous manuscript, which is indeed hard to tell the benefits of the method. In the revised manuscript, we further interpreted the improved gravity increase method as "This method leads to the failure of a slope in PFC2D by slowly increasing gravity acceleration and reducing the friction coefficient of particles while keeping other parameters constant. Notably, the amplitude of reduction of the friction coefficient is the same as that of the increase in gravity acceleration. In this way, the resisting force can be fixed and therefore the factor of safety is directly reflected by the driving force". (Page 8: Lines 244-247)

Comment 23: Line 202: this seems a very stable slope. Expectedly, in view of the amount of rock bridges along the rupture surface, which may be estimated at about 30-40%, according to figure 7. Response: Thank you very much for your comment. It is really true that the investigated rock slope is extremely stable, which can be reflected by the high factors of safety. As you said, the amount of rock bridges along the rupture surface can also verify that the investigated slope is very stable.

*Comment 24:* Line 232 (and after): Perhaps it will be better to refer to the "slip mass" as "failed", or "detached" mass/volume/material.

**Response**: Thank you very much for your suggestion. It is really true that "failed mass" is better than "slip mass"; thus, we carefully checked all the manuscript and changed "slip mass" to "failed mass". (Page 9: Lines 282-283, 284; Page 10: Lines 288-289; Page 11: Lines 327-328)

*Comment 25:* Line 255: I recommend referring to the "model geometry" or "morphology", rather than "shape".

**Response:** Thank you very much for your suggestion. It is really true that "morphology" is better than "shape". We replaced "shape" with "morphology" in the revised manuscript. (Page 10: Line

*Comment 26:* Line 272: A variability 25-75 is indeed very high. Perhaps this variability be lower if a more realistic DFN (i.e. inclusive of smaller fractures) was to be simulated. Absolute values would be lower, for sure.

**Response**: Thank you very much for your comment. It is really true that the variability between factors of safety is very high. In the revised manuscript, we reduced the strength of intact materials to account for the smaller fractures, which has been explained in the response to Comment 1. In the recalculation of factors of safety, a lower variability was indeed observed (from 12 to 38) and absolute values were also lower. The result showed that the final factor of safety of the rock slope is 19, which is lower than the previous one (43.5). (Page 24: Figure 12a)

Comment 27: Line 290: again, just a comment. The limitations of this estimation is that is assumes that the base of the model is constituted by strong rock, likely with high coefficient of restitution, and the distribution of the failed mass over this distance is not considered.

**Response**: Thank you very much for you comment. It is really true that the base of the model is constituted by strong rock, which is represented by a rough rigid wall in PFC. The distribution of the failed mass over the distance is not analysed since this result cannot been proved a statistical significance. The accumulation results vary for 100 different SRM models, which can be verified in Fig. 13. The only thing common is that the final deposit is composed of relatively intact rock blocks and crushed particles, and the blocks pile up above the crushed particles, presenting an inverse grading phenomenon.

**Comment 28:* Lines 301-303: this sentence is unclear and requires rephrasing.**

**Response**: We are very sorry for our unclear description. In the revised manuscript, we rewrote this sentence as "The factor of safety of the investigated slope is extremely high but reasonable. In the field investigation, weak interlayer and through-going discontinuities are not observed. The non-persistent fractures are very developed, which therefore play a vitally important role in the stability of the investigated slope". (Page 12: Lines 352-354)

*Comment 29:* Lines 303-305: I agree with the authors here: rock bridges are multiple orders of magnitude stronger than discontinuities, and this justifies the high FoS. The questions, however, is:

306)

are these estimations accurate and representative of the real situation? Figure 4 shows a rock mass significantly more fractured that the DFNs employed in this study, where the slope is formed by very large, intact blocks.

**Response**: Thank you very much for your comment. On the basis of our previous results (small fractures are not considered), the factors of safety are accurate and can represent the real situation. This is because we also calculate the factors of safety by the traditional methods (i.e., the ratio of the resisting force to the driving force), which presents the same result as the simulation ones.

The DFNs in Fig. 4 are totally introduced into the simulated slopes, which is reflected by the comparison between Fig. 4 and Fig.7. As for very large and intact blocks you mentioned, maybe you refer to the blocks of the boundary sections located in the bottom and right sides of the slope section. The boundary section won't affect the slope stability, which mainly contributes to overcome boundary effect. In the revised manuscript, we added the description regarding the boundary section as "The bottom and right sides of the slope section were expanded by 10 m as the boundary section, which aims to avoid boundary effect and does not affect the slope stability (Fan et al., 2004)". (Page 7: Lines 220-222)

**Comment 30:** Figure 1c: a scale and possibly a north arrow is required**

**Response**: Thank you very much for your suggestion. In the revised manuscript, we added the scale and the strike of the slope in Fig. 1c. (Page 17: Figure 1c)

*Comment 31:* Figure 7: a legend bar (stress) and scales are needed for clarity

**Response:** Thank you very much for your suggestion. In the revised manuscript, we added the legend bar and scales in Fig. 7 as you suggested. (Page 20: Figure 7)

*Comment 32:* Figures 8 and 10: the use of an uniform color bar and legend would enhance the comparison of the states depicted by each sub-figure.

**Response**: Thank you very much for suggestion. It is really true that a uniform color bar and legend makes it convenient to compare different stages in each picture. However, the selected stages show considerable differences of displacements (especially stages in Fig. 8), a uniform color bar and legend presents potential drawbacks. If a small color bar and legend is adopted, the large displacements will appear the same color (red here), which makes it difficult to distinguish

the displacements of different rock blocks; if a large color bar and legend is used, the initial displacements is difficult to be captured and cannot make the profile of slip surface clear. Therefore, we are afraid that a uniform color bar and legend cannot be realised in Figs. 8 and 10. Comment 33: Table 4: I recommend using the same order for micro, numerical, and lab

parameters: friction, normal, and tangential stiffness.

**Response**: Thank you very much for your suggestion. The same order for micro, numerical and lab parameters is more beneficial for comparing the results of parameter determination. In the revised manuscript, we changed the order of parameters to ensure they are orderly arranged. (Page 25: Table 4)

We thank you for your valuable comments and suggestions. These comments are all valuable and helpful in revising and improving our paper, as well as in guiding the significance of our research.

Relevant references:

Bonilla-Sierra, V., Scholtès, L., Donzé, F. V., and Elmouttie, M. K.: Rock slope stability analysis using photogrammetric data and DFN–DEM modeling, Acta Geotech., 10, 497–511, 2015.

Chen, J. P., Shi, B. F., and Wang, Q.: Study on the dominant orientations of random fractures of fractured rock mass, Chinese Journal of Rock Mechanics and Engineering, 24, 241–245, 2005.

Damjanac, B., Cundall, P.: Application of distinct element methods to simulation of hydraulic fracturing in naturally fractured reservoirs, Comput. Geotech., 71, 283-294, 2016.

Elmo, D., Stead, D., Eberhardt, E., and Vyazmensky, A.: Applications of finite/discrete element modeling to rock engineering problems, Int. J. Geomech., 13, 565–580, 2013.

Fan, S. C., Jiao, Y. Y., and Zhao, J.: On modeling of incident boundary for wave propagation in jointed rock masses using discrete element method, Comput. Geotech., 31, 57–66, 2004.

Kulatilake, P. H. S. W. and Wu, T. H.: Estimation of mean trace length of discontinuities, Rock Mech. Rock Eng., 17, 215–232, 1984.

Lajtai, E.Z.: Shear strength of weakness planes in rock, Int. J. Rock Mech. Min., 6, 499-515,

1969b.

Lorig, L. J., Darcel, C., Damjanac, B., Pierce, M., and Billaux, D.: Application of discrete fracture networks in mining and civil geomechanics, Mining Technology, 124, 239-254, 2015.

Priest, S. D.: Discontinuity analysis for rock engineering, London : Chapman and Hall, 1993.

Pine, R. J., Coggan, J. S., Flynn, Z. N., and Elmo, D.: The development of a new numerical modelling approach for naturally fractured rock masses, Rock Mech. Rock Eng., 39, 395–419, 2006.

Shang, J., West, L. J., Hencher, S. R., Zhao, Z.: Geological discontinuity persistence: Implications and quantification, Eng. Geol., 241, 41-54, 2018.

Terzaghi, K.: Stability of steep slopes on hard unweathered rock, G éotechnique, 12, 251–270, 1962.

Zhang, W., Zhao, Q. H., Chen, J. P., Huang, R. Q., and Yuan, X. Q.: Determining the critical slip surface of a fractured rock slope considering preexisting fractures and statistical methodology, Landslides, 14, 1253–1263, 2017.

Zhang, W., Fu, R., Tan, C., Ma, Z. F., Zhang, Y., Song, S. Y., Xu, P. H., Wang, S. N., Zhao, Y. P.: Two-dimensional discrepancies in fracture geometric factors and connectivity between field-collected and stochastically modeled DFNs: A case study of sluice foundation rock mass in Datengxia, China, Rock Mech. Rock Eng., 53, 2399–2417, 2020a.

Zhang, W., Lan, Z. G., Ma, Z. F., Tan, C., Que, J. S., Wang, F. Y., Cao, C.: Determination of statistical discontinuity persistence for a rock mass characterized by non-persistent fractures. Int. J. Rock Mech. Min., 126, 104177, 2020b.

**Stability evaluation and potential failure process of rock slopes characterized by non-persistent fractures**

Wen Zhang1, Jia Wang1, Peihua Xu1, Junqing Lou2, Bo Shan2, Fengyan Wang3, Chen Cao1, Xiaoxue Chen1, Jinsheng Que4

[revised manuscript text omitted]

---

## Author Response (AR2)

Dear Editor and Referee:

Thank you very much for your valuable comments and suggestions concerning our manuscript entitled "Stability evaluation and potential failure process of rock slopes characterized by non-persistent fractures" (ID: nhess-2020-58). These comments are all valuable and helpful for revising and improving our paper, as well as the important guiding significance to our researches. Careful revisions have been made to the manuscript according to the comments and suggestions, which we hope will merit your approval. The revised portions are marked in red in the manuscript. A point-by-point response to your comments and relative changes made in the manuscript are listed as follows.

**Responses to Editor**

*Comment 1:* after your first revision, the manuscript seems much improved and most of the issues raised by the reviewers have been successfully addressed. However, there remains one major concern regarding the mechanical parameters and the modelization of the smaller discontinuities not included in the DFN. According to the referee's belief, to which I tend to agree, the issue must be solved before publication is possible.

*Response:* Thank you very much for your positive evaluation regarding to our last modification. It is realty true that we did not deal well with the effect of smaller fractures on mechanical parameters of the rock mass between the large fractures. Referee #1 pointed that the linear persistence we used essentially depended on the DFN, which was the main reason that Referee #1 had questions about our method. We have to acknowledge that the description of linear persistence and macro-properties of the rock mass between the large fractures is indeed unclear. The linear persistence we used only considers the rock mass between the large fractures. There are only small fractures (fractures with trace lengths smaller than 1.5 m) in the rock mass between the large fractures; thus, the value of linear persistence is the ratio of fracture trace lengths (only the trace lengths of small fractures) and the total length of coplanar given line. From the analysis above, the linear persistence is controlled by small fractures rather than the DFN (the large fractures). In the

revised manuscript, we rewrote this section and further explained that the parameters required in this method depend on the rock mass itself (including small fractures and intact rock material) and have nothing to do with the large fractures (DFN). Relative changes in the revised manuscript have been listed in the responses to Referee #1 and therefore are not described in detail in this place. As for the approach proposed by Referee #1, i.e., estimating the strength of rock material by an estimation of the GSI of the rock mass between the large fractures, we tried to calculate four slope models by this approach, and the simulation results are similar to those obtained by the method we used. Considering the large number of slope models (100 models) and the small difference between the results of the two methods, we remained the previous method.

***Comment 2:*** Other than this, there are still several minor issues that are highlighted singularly in the following Referee #1 report. Please respond to them one by one and try to modify the paper accordingly.

***Response:*** Thank you very much for your comment. We have carefully looked at and replied to all comments and suggestions one by one, which can be found in Responses to Referee #1. In addition, the previous manuscript have been modified according to these comments and suggestions, which can be found in the relative responses as well as the revised manuscript.

***Comment 3:*** I am confident that, after those issues will be cleared, the manuscript will become acceptable.

***Response:*** Thank you very much for your comment. Careful revisions have been made to the manuscript according to the comments and suggestions, which we believe that all issues has been cleared. We really hope that our revised manuscript will merit your approval.

***Comment 4:*** I will personally review your changes for this second round so that no further peer-review report will be needed. This should expedite publication in case you are willing to modify the paper following our suggestions.

***Response:*** Thank you very much for your hard work. We tried our best to improve and make changes to the manuscript following your professional suggestions. The revised portions are marked in red in the revised manuscript. Meanwhile, relative revisions are also listed in the corresponding responses to Referee #1.

Once again, thank you very much for your professional comments and suggestions! We sincerely appreciate your work and hope that our revised manuscript will be met with approval.

**Responses to Referee #1**

**Comment 1:** In this reviewed version of the manuscript, most of the points raised in the previous review have been addressed. The paper still requires some polishing here and there (including in writing quality, see point-by-point comments), but overall clarity improved. There is still one point of concern, again related to the estimation of the strength of the equivalent rock mass, which does not sound very solid. Specifically, the Authors used a failure criterion originally developed for non-persistent discontinuities, rather than rock masses. Importantly, however, in the presented approach the equivalent rock mass strength is dependent on the DFN already included in the model, and not the smaller discontinuities that were not included in the DFN. This means that although the factor of safety is lower, as expected, because the intact material is weaker than in the previous simulation, the mechanical properties responsible for such a decrease seem rather arbitrary. A more solid method would comprise the estimation of the GSI (excluding the fractures included in the DFN) to derive the equivalent shear strength parameters for the rock mass between larger discontinuities. In this reviewer's opinion this point cannot be overlooked, and it is critical that the Authors address it adequately. Also, some tables appear not to have been updated with the new properties used for the simulations.

**Response:** Thank you very much for your positive comments on our last modification. It is really true that the manuscript needs to be further revised and supplemented, which we has carefully looked at and modified according to your line-by-line comments.

The estimation of the rock mass between the large fractures applied a failure criterion which was developed by Lajtai (1969b) and Shang et al. (2018) based on the Mohr-Coulomb criterion. It is really true that this failure criterion is originally used for the calculation of the shear strength of non-persistent fractures. Non-persistent fractures in this criterion refer to the combination of rock bridges and persistent fractures along the shearing direction of the rock mass, which means non-persistent fractures is actually the shear failure surface of the rock mass. Therefore, it is

reasonable that this failure criterion is applied to approximately estimate the shear strength of the rock mass between the large fractures.

When estimating the shear strength (specifically, the cohesion and friction angle) of the rock mass between the large fractures, we only focus on the rock mass containing the small fractures which are not collected in the field. Therefore, the shear strength of the rock mass is controlled by small fractures and intact rock material rather than DFN. BPM parameters are ascertained and quantified based on the macro-properties of the rock mass between the large fractures. Therefore, the decrease of mechanical properties is not arbitrary. It is really true that the description of this section is unclear which may make readers confused. In the revised manuscript, we rewrote this section to ensure the issues described clearly.

The method you suggested, i.e., deriving the shear strength of the rock mass between the large fractures by the estimation of the GSI, is really worthwhile to try. We selected four slope models to conduct the stability analysis based on the parameters obtained by the estimation of the GSI. The results are similar to those obtained by the method we used. Given the large amount of time required to calculate all the models (100 models) and the tiny difference between the results obtained by the two methods, we still adopt the original method.

In the revised manuscript, we rewrote the relative contents and further explained that we only considered the rock mass between the large fractures. Details can be found in Lines 159-193 in the revised manuscript, which are also listed in the end of this response. In addition, we are very sorry for our carelessness. In the revised manuscript, Tables 2 and 3 have been updated with the new properties used for the simulations. (Page 26: Tables 2 and 3).

[revised manuscript text omitted]

**_Comment 2:_** Lines 63-64: please revise this sentence, as it seems that only model is investigated (i.e., Authors state that "a single-slope model" is analyzed). And, if that is actually the case, it is unclear how the runout analysis was undertaken for all the DFN realizations.

**_Response:_** Thank you very much for your suggestion. We investigated all slope models generated with all the DFN realizations (100 DFNs); thus, "a single-slope model" described here is indeed inappropriate. In the revised manuscript, we corrected this sentence as "Third, the improved gravity increase method is employed to determine the stability of all slope models generated with 100 DFNs". (Page 2: Lines 63-64)

**_Comment 3:_** Line 71: please change "became" with "develop"

**_Response:_** Thank you very much for your suggestion. In the revised manuscript, the word "became" has been changed to "develop". (Page 3: Line 71)

**_Comment 4:_** Line 72: I suggest using "Understanding whether rock slides will happen requires adequate/careful geological investigations". Simply stating "calculating and evaluating" leaves the reader wondering what should be calculated and evaluated.

**_Response:_** Thank you very much for your suggestion. It is really true that stating "calculating and evaluating" makes readers questionable. In the revised manuscript, we have used "Understanding whether rock slides will happen requires careful geological investigations" to substitute "Whether rock slide will happen requires calculating and evaluating". (Page 3: Lines 72-73)

**_Comment 5:_** Line 74: I recommend using "by a low-mountain terrain, characterized by higher elevation to the north than to the south"

*Response:* Thank you very much for your suggestion. In the revised manuscript, we rewrote this sentence as you suggested, i.e., "The Laohuding Quarry area is in a low-mountain terrain, characterized by higher elevation to the north than to the south". (Page 3: Line 74)

*Comment 6:* Lines 84-85: probably bedding planes and weak interlayers do not provide useful information with regard to tectonic stability. Also, I assume that this statement is backed by historic seismicity data, which is probably more indicative than the presence of faults and faults.

*Response:* Thank you very much for your comment. It is really true that bedding plane and weak interlayers cannot serve as the indicators of tectonic stability. In addition, the tectonic stability is indeed evaluated by historic seismicity data, as well as the presence (or absence) of faults and shear zones. In the revised manuscript, we explained this sentence further as "Bedding planes, weak interlayers, faults, folds and shear zones are not observed. This area is tectonically stable according to historic seismicity data". (Page 3: Lines 84-85)

*Comment 7:* Line 101: Note that the K value is not a constant. I recommend changing in "The computed Fisher value (K) for the various sets……imply an high dispersion…".

*Response:* Thank you very much for your suggestion. It is really true that $K$ value is not a constant and different $K$ values exist in various fracture sets. In the revised manuscript, we rewrote the description regarding to $K$ values as "The computed Fisher constants ($K$) for three fracture sets are 17.1, 10.3, and 9.1, which imply the high dispersion of fracture orientations (Fisher, 1953; Priest, 1993)". (Page 4: Lines 101-102)

*Comment 8:* Lines 103-105: the Authors state the different "possibilities" that may result in bias, but do not explain how such bias is generated (ie, the window is can be smaller than the discontinuities, and the actual trace length cannot be measured). It is fair to say that this is basic knowledge, however, the Authors write that "the measures trace length bias occurs due to…". Therefore, the reader would expect a more detailed explanation is to be expected in the following.

*Response:* Thank you very much for your suggestion. We indeed ignored the basic reason why the measured trace lengths bias occur. In the revised manuscript, we explained trace length as "The measured trace lengths bias occurs due to the following: the size of the sampling window is usually smaller than that of the outcrop. When the trace length is larger than the sampling window,

the actual trace length cannot be totally measured, i.e., only the part of the trace length inside the sampling window can be measured. Specifically, the measured results can be divided into three types: a) only one end of a fracture trace is measured, (b) both ends of a fracture trace are measured, and (c) no end of a fracture is measured". (Page 4: Lines 103-108)

***Comment 9:*** Line 115: change "from the exposed surface" to "across the exposed surface".

***Response:*** Thank you very much for your suggestion. In the revised manuscript, we has changed "from the exposed surface" to "across the exposed surface". (Page 4: Line 118)

***Comment 10:*** Line 133: unclear what "fracture data" refers to – perhaps are the Authors referring to "fracture intensity"?

***Response:*** we are very sorry for unclear description. It is really true that fracture intensity belongs to the statistical fracture data. Besides, statistical fracture data include the mean and variance values of the trace length square, the distribution types of fracture orientation (empirical distribution) and location (Possion's distribution). In the revised manuscript, we added detailed description as "More than one DFN can be generated on the basis of the aforementioned statistical fracture data, including the distribution type of fracture locations (Possion's distribution), 2D fracture intensity ($P_{21}$), the distribution type of fracture orientation (empirical distribution), and the mean [$E(ch^2)$] and variance [$V(ch^2)$] values of the trace length square". (Page 5: Lines 135-137)

***Comment 11:*** Lines 136-139: this part remains unclear. "We intersected the fractures….along the dip direction". Did the Authors virtually extend the fractures across the section, to identify the points where these extended fractures would intersect one another (if this is correct, the term "exposed surface" is ambiguous)? The subsequent procedure is still not clear – I suggest including a conceptual figure to enhance clarity.

***Response:*** We are very sorry for unclear description. We did not extend fractures, instead, we left them as they are. The intersections are obtained by the intersection of original fractures with lines we set in the exposed surface. In the revised manuscript, we rewrote the description of this procedure as "To solve this problem, we generated and verified the validity of numerous DFN models by applying the procedure in Fig. 5. Initially, we set lines along the strike of the exposed surface and fractures in the exposed surface can be intersected with these lines. A series of

intersection points can be obtained for an individual line. Subsequently, the same procedure was performed to the generated DFNs, i.e., we intersected the DFNs using lines along the strike of the cross section and consequently generated a series of intersection points for an individual network. We compared these two sets of intersection points by using their probability density curves. When the results were identical to one another, the DFNs generated in the cross section were proved reasonable and can be selected. Finally, we selected 100 reasonable DFN models to construct different slope models and conduct stability analysis and potential failure process simulation". (Page 5: Lines 140-146)

In addition, we added a conceptual figure to help readers understanding the procedure as you suggested (Figure 5). (Page 19: Figure 5)

Notably, the order of Figures 5-12 has been changed to Figures 6-13 in the revised manuscript due to the addition of new Figure 5.

**_Comment 12:_** Lines 159-160: I suggest the Authors to refer to the "trial-and-error method" to describe the procedure of repeatedly changing particle size to achieve the best compromise between runtime and detail.

**_Response:_** Thank you very much for your suggestion. It is really true the procedure of repeatedly changing particle size to achieve the best compromise between runtime and detail is consistent with trial-and error method. In the revised manuscript, we referred to the trial-and-error method and described the procedure as "In the present study, the trial-and-error method was used to find a best balance of reasonable results and high computational efficiency. Specifically, we repeatedly changed the particle sizes and checked the simulation result and the time took. Particles with radii between 0.05 and 0.083 m were finally selected to fill rock specimens". (Page 7: Lines 200-202)

**_Comment 13:_** Line 181: In the present form, there is a mixture between actual geological processes (progressive strength deterioration) and the numerical approach used to compute the factor of safety (the SSR). I recommend the Authors to modify this sentence, either changing the brackets with "which can be investigated using the shear strength reduction method" or stating that "the rock mass failure is generally SIMULATED/INVESTIGATED by reducing the shear strength".

*Response:* Thank you very much for your suggestion. It is really true that there is a mixture between actual geological processes and the numerical approach used to compute the factor of safety. In the revised manuscript, this sentence has been deleted because we rewrote the whole section regarding to the estimation of the macro-properties of the rock mass between the large fractures.

*Comment 14:* Line 188: Units, ° and MPa, should be included after the values. Please correct "filed" in "field".

*Response:* We are very sorry for our carelessness. It is really true that the units of relative parameters should be included after the parameter values. In the revised manuscript, we has added the units of parameters, such as 12.25 MPa, 25°, 18°, 6.125 MPa and 21.58°. (Page 6: Lines 183 and 192. In addition, "filed" has been changed to "field" in the revised manuscript. (Page 6: Line 183)

*Comment 15:* Lines 183-193: The procedure for computing the linear persistence and the rock mass strength is unclear. The Authors use the equation from Shang et al., 2018 (which is used to estimate the strength of non-persistent discontinuities with co-planar rock bridges) in order to evaluate the strength of the equivalent rock mass. More precisely, they are effectively estimating the shear strength along the rupture surface. Authors refer to cohesion and friction angle of "intact rock mass". This should be simply "intact rock", as the equivalent rock mass strength is being estimated. In any case, the equation simply involves a balance between the strength of intact rock and fractures, as a function of the linear persistence of the latter. In other words, a persistence of 50% implies that the strength is given for a 50% by the fractures, and for the other 50% by the intact rock. Here is the problem: in the equation, the Authors used the strength (cohesion and friction angle) of the intact rock and fractures to compute an equivalent rock mass strength. As such, the value obtained is a rough estimation of the strength of an equivalent continuum rock mass constituted by DFN + intact rock. Then, this strength is assigned to the intact material in the model, while leaving the DFN in place. It is true that this approach results in a decrease in the FoS of the slope, simulating, in a way, the presence of a fractured rock mass between DFN fractures. On the other hand, such a decrease in strength is not based on actual field data (ie, small fractures),

but only on the value of linear persistence, which essentially depends on the DFN. A more sensible approach would be an estimation of the GSI of the rock mass between the large fractures, in order to estimate its strength, and assign it to the intact material.

*Response:* Thank you very much for your comment. It is really true that the description of the procedure for computing the linear persistence and the rock mass strength is unclear, which is obvious according to your comment. The equation from Shang et al., 2018 is originally used for the calculation of the shear strength along the rupture surface. The shear failure of the rock mass generally occurs along the rupture surface. Therefore, it is reasonable that this equation is used for approximately estimating the shear strength of the rock mass between the large fractures.

Your description of this equation is basically right, i.e., the shear strength (cohesion and friction angle) of the rock mass is estimated according to that of the fractures and intact rock material. The most important reason you question the method we used is that "a decrease in strength is not based on actual field data (i.e., small fractures), but only on the value of linear persistence, which essentially depends on the DFN." We have to acknowledge that the unclear description leads to misunderstanding. The estimation of the shear strength of the rock mass only focused on the rock mass between the large fractures (the fractures considered in the DFN). Therefore, the value of linear persistence is controlled by the small fractures in the rock mass (excluding the fractures included in the DFN). In addition, the linear persistence is measured by setting lines in the rock mass between the large fractures in the field, which is actual field data.

In the revised manuscript, we rewrote the relative contents to ensure the issues described clearly. Revised portions can be found in the response to Comment 1.

As for the method you suggested, i.e., deriving the shear strength of the rock mass between the large fractures by the estimation of the GSI, is really worthwhile to try. We selected four slope models to conduct the stability analysis based on the parameters obtained by the estimation of the GSI. The results are the similar to those obtained by the method we used. Given the large amount of time required to calculate all the models (100 models) and the small difference between the results obtained by the two methods, we still adopt the original method.

*Comment 16:* Line 202: please remove "by" from "by this way".

*Response:* Thank you very much for your suggestion. In the revised manuscript, we rewrote this section and "by this way" has been deleted.

*Comment 17:* Line 203: please change "equivalent parameters" with "microparameters".

*Response:* Thank you very much for your suggestion. In the revised manuscript, we rewrote this section and "equivalent parameters" has been deleted.

*Comment 18:* Line 205: please change "generate" with "simulate".

*Response:* Thank you very much for your suggestion. In the revised manuscript, we has changed "generate" to "simulate". (Page 7: Line 213)

*Comment 19:* Figure 1: I recommend simply using a North arrow (just with the N) and state the strike in the caption. Otherwise it is unclear what exactly the 200 stands for.

*Response:* Thank you very much for your suggestion. In the revised manuscript, we deleted the arrow representing the strike of the slope and substituted it with a North arrow. In addition, we stated the strike of the slope in the caption as you suggested. (Page 17: Figure 1)

*Comment 20:* Figure 2: many fractures are trimmed by the window. Check if long vertical fractures are included in the DFN.

*Response:* Thank you very much for your comment. It is really true that many fractures are trimmed by the window which is inevitable for the sampling window method. In the manuscript, we has corrected the bias caused by trimming, which were described in Section 2.2. We checked 2D trace map and ensured that all fractures we collected were included in the DFN.

*Comment 21:* Figure 3: please include the number of poles plotted in each stereonet.

*Response:* Thank you very much for your suggestion. In the revised manuscript, we added the number of poles in each stereonet (i.e., 18, 58, and 93 poles, respectively). (Page 18: Figure 3)

*Comment 22:* Tables 2-3: the micro-parameter values were not updated, with respect with the previous version of the manuscript, to reflect the use of the equivalent rock mass.

*Response:* we are very sorry for carelessness. In the revised manuscript, we updated the parameters in Tables 2 and 3 to reflect the properties of the rock mass between the large fractures. (Page 26: Tables 2 and 3)

We thank you for your valuable comments and suggestions. These comments are all valuable and helpful in revising and improving our paper, as well as in guiding the significance of our research.

[revised manuscript text omitted]

---

## Author Response (AR3)

Dear Editor:

Thank you very much for your hard work concerning our manuscript entitled "Stability evaluation and potential failure process of rock slopes characterized by non-persistent fractures" (ID: nhess-2020-58).

We are very sorry for missing one financial support in the manuscript, which have been added in the text (.docx) with red marked (Page 13: Line 413 and Page 14: Line 417). This modification is not technical but only informative, which has met the handling editor (Filippo Catani)'s approval.

Once again! We sincerely appreciate your work and are really happy that our manuscript was accepted for final publication in NHESS.